



# TICOI: an operational Python package to generate regularized glacier velocity time series

Laurane Charrier[1], Amaury Dehecq[1], Lei Guo[1,2], Fanny Brun[1], Romain Millan[1], Nathan Lioret[1], Luke Copland[3], Nathan Maier[4], Christine Dow[5], and Paul Halas[6]

[1]Institut des Géosciences de l'Environnement, Université Grenoble-Alpes, Grenoble, France
[2]School of Geo-science and Info-physics, Central South University, Changsha, China.
[3]Department of Geography, Environment and Geomatics, University of Ottawa, Canada
[4]Los Alamos National Laboratory, Los Alamos, New Mexico
[5]Department of Geography and Environmental Management, University of Waterloo, Canada
[6]Independant Researcher formerly at University of Bergen, Bjerknes Centre for Climate Research, Bergen, Norway

**Correspondence:** Laurane Charrier (laurane.charrier@univ-grenoble-alpes.fr)

**Abstract.**

Ice velocity is a crucial observation as it controls glacier mass redistribution and future geometry. While glacier annual velocities are now available in open-source worldwide, sub-annual velocity time series are still highly uncertain and available at heterogeneous temporal resolutions. This hinders our ability to understand flow processes, such as basal sliding or surges, and integration of these observations in numerical models. We introduce an open source and operational Python package called TICOI (Temporal Inversion using Combination of Observations and Interpolation). TICOI fuses multi-temporal and multi-sensor image-pair velocities produced by different processing chains, using the temporal closure principle. In this article, we provide extensive examples of TICOI application on the ITS_LIVE dataset and in-house velocity products. The results are validated using GNSS data collected on three glaciers with different dynamics in Yukon and western Greenland, including a surging glacier. Comparison with GNSS observations demonstrates a reduction in error by up to 50% in comparison with the raw image-pair velocities and other post-processing methods. This increase in performance comes from the development of methodological strategies to enhance TICOI's robustness to temporal decorrelation and abrupt non-linear changes. TICOI also proves to be able to retrieve monthly velocity when only annual image-pair velocities are available. This package opens the door to the regularization of various datasets, enabling the creation of standardized sub-annual velocity products.

## 1 Introduction

Glacier surface velocity monitoring is key to understand how glaciers are changing in a warming climate. Global glacier thinning (Hugonnet et al., 2021) is expected to lead to large changes in ice flow, which in turns affect mass redistribution and glacier geometry (Dehecq et al., 2019). Documenting ice velocity and changes in ice velocity is thus important to understand



the future evolution of glaciers. Regional to global mapping of glacier surface velocity from remote sensing helped to constrain estimates of glacier thickness (Millan et al., 2022) and ice fluxes into the ocean (Gardner et al., 2018; Mouginot et al., 2019; Kochtitzky et al., 2022), which led to reductions on present and future glacier contribution to sea level rise.

  While these applications rely on multi-annual velocity averages, many glacier processes should be studied at sub-annual scales, in particular to study calving events (Provost et al., 2024; Riel et al., 2021), glacier response to lake drainage (Maier

et al., 2023; Main et al., 2023; Wendleder et al., 2023), surface runoff (Wendleder et al., 2023), changes in the efficiency of subglacial hydrological networks (Nanni et al., 2023; Maier et al., 2023) or glacier surges (Copland et al., 2011; Quincey et al., 2015; Beaud et al., 2021). Additionally, many recent methodological and modelling developments stress the need for precise and temporally resolved velocity products to infer basal conditions (Jay-Allemand et al., 2011; Goldberg et al., 2015) or near-future projection (Choi et al., 2023), for example using transient inverse methods (Goldberg et al., 2015; Choi et al.,

30   2023).

  So called image-pair velocities are derived by calculating the displacement of features between two images. Recent improvements in satellite image quality and resolution, with the launch of Landsat-8 in 2013, Sentinel-1 in 2014 and Sentinel-2 in 2015, amongst others, have made it possible to derive image-pair velocities, with an enhanced signal-to-noise ratio at a relative high frequency (5 to 16 days at the best). Despite the recent increase in available velocity datasets, post-processed time series of

sub-annual glacier velocities, sampled on fixed time intervals, are not yet available at a global scale. Only individual image-pair velocities have been released, such as those from the ITS_LIVE project, which provides velocities derived from correlating Landsat-4,5,6,7,8, and Sentinel-1,2 image pairs, separated by temporal baselines of 5 to 400 days. These products remain noisy and sparse, especially over mountain glaciers. Moreover, image-pair velocities are difficult to interpret because they contain velocities measured using image pairs from different satellites, with different temporal baselines. For these reasons, many

research teams processed their own image-pair velocities instead of using existing datasets (Yang et al., 2022; Nanni et al., 2023; Wallis et al., 2023; Derkacheva et al., 2020; Halas et al., 2023; Provost et al., 2024; Beaud et al., 2021), to reach a better signal-to-noise ratio and improve interpretability. However, this requires high computational and storage resources, and often suffers from a lack of reproducibility. For the same reason, image-pair velocities with variable dates are difficult to include into models, whereas seasonal velocity could aid in retrieving basal conditions (Derkacheva et al., 2021) or ice rheology properties

(Bolibar et al., 2023). This calls for a standardized framework dedicated to the processing of available image-pair velocity products to produce consistent sub-annual time series of glacier velocity.

  Several methods have been proposed to produce velocity time-series: cubic spline or LOWESS regression (Derkacheva et al., 2020), regression using a dictionary of B-splines (Riel et al., 2021), sinusoidal regression (Greene et al., 2020), Bayesian recursive smoother (Wallis et al., 2023) or temporal closure (Altena et al., 2019; Charrier et al., 2022b). However, most of

these methods use only a subset of the available data, such as velocities quantified using small temporal baselines (<100 days) (Derkacheva et al., 2020; Riel et al., 2021; Nanni et al., 2023; Wallis et al., 2023), or use strong assumptions about glacier behavior, such as sinusoidal variations in seasonal motion (Greene et al., 2020). Moreover, even though efforts have been undertaken to compare some of the image correlation algorithms (Zheng et al., 2023; Heid and Kääb, 2012; Jawak et al.,





2018), no study has tried to fuse datasets produced by different research teams to obtain sub-annual time-series. Finally, few
of these methods are open-source.

The objective of this study is to propose an open-source and operational package able to fuse image-pair velocities computed
using different satellite images, with different temporal baselines, and possibly using different processing chains, in order to
obtain regularized (i.e., sampled at regular time steps) velocity time-series with an associated quality indicator. To do this,
we rely on a method based on the temporal closure of the displacement network, also known as a Small BAseline Subset
(SBAS)-like approach, which originated from the Interferometric Synthetic Aperture Radar (InSAR) community (Berardino
et al., 2002; Doin et al., 2011). These approaches have previously been adapted and applied to glaciers to retrieve 2D or
3D velocity time-series (Charrier et al., 2022b, a; Guo et al., 2020; Samsonov et al., 2021), but have not been applied in an
operational framework because: 1) they remain sensitive to temporal decorrelation (*i.e.*, very low velocity values which can
be measured when strong surface changes occur, or near the margins of the moving object. This phenomenon is especially
visible when the temporal baseline is large and/or when the velocity is low.); 2) they often include a regularization term which
assumes acceleration to be zero in time (which is not true for surging glaciers or glaciers with a strong seasonality pattern), or
a pre-defined mathematical model (i.e. a periodical model) which requires *a priori* knowledge on the glacier dynamic; 3) they
have never been evaluated against global navigation satellite system (GNSS) data; and 4) computational costs are high.

To overcome these issues, we present a new method called Temporal Inversion using Combination of Observations and
Interpolation (TICOI) to derive regularized glacer velocity time series. First, we detail our methodological strategy to increase
the robustness of previous developments against temporal decorrelation and abrupt non-linear changes. We also propose three
criteria to evaluate the quality of our results, including an error propagation, that is based on a robust theoretical framework.
Then, we validate this new approach against seven GNSS (Global Navigation Satellite System) stations located on three differ-
ent glaciers, including one with an active surge phase, and two others showing seasonal variations. We carry out a sensitivity
analysis, and illustrate our results along glacier center flowlines. Then, we discuss the interest of fusing datasets originating
from different processing chains and the possibility to retrieve sub-annual velocity time-series from annual image-pair veloci-
ties only. Finally, we discuss the potential application of TICOI at a regional and global scale.

## 2 Method

In this section, we describe the TICOI workflow (Fig. 1). The first part builds on previous developments (Charrier et al.,
2022b, a), the second part presents methodological strategies to improve the computational performance and robustness of
the method to abrupt non- linear changes in surface velocities and temporal decorrelation. The aim is to provide a robust
post-processing package that does not require *a priori* knowledge of the ice flow behavior.



## 2.1 Previous developments

### 2.1.1 Temporal closure's principle

The temporal inversion is based on the temporal closure of the displacement measurement network (Berardino et al., 2002). Temporal closure links $n$ measured displacements in $Y$ to $p$ estimated displacements in $X$ (Fig. 1), by a system of linear equations. This system of linear equations can also be written as $AX = Y$, with $A$ the design matrix linking $X$ to $Y$. To understand the structure of $A$, let's take an example with three displacements represented in Figure 1. Assuming that the displacement is cumulative in time, it can be written that:

$$\begin{cases} d_{t_0,t_6} = \hat{d}_{t_0,t_4} + \hat{d}_{t_4,t_5} + \hat{d}_{t_5,t_6} \\ d_{t_0,t_4} = \hat{d}_{t_0,t_4} \\ d_{t_4,t_6} = \hat{d}_{t_4,t_5} + \hat{d}_{t_5,t_6} \end{cases} \tag{1}$$

with $d_{t_i,t_j}$ a measured displacement between dates $t_i$ and $t_j$, and $\hat{d}_{t_k,t_l}$ a displacement estimated between dates $t_k$ and $t_l$.

This is equivalent to $AX = Y$, which is in this example:

$$\begin{bmatrix} 1 & 1 & 1 \\ 1 & 0 & 0 \\ 0 & 1 & 1 \end{bmatrix} \begin{bmatrix} \hat{d}_{t_0,t_4} \\ \hat{d}_{t_4,t_5} \\ \hat{d}_{t_5,t_6} \end{bmatrix} = \begin{bmatrix} d_{t_0,t_6} \\ d_{t_0,t_4} \\ d_{t_4,t_6} \end{bmatrix} \tag{2}$$

It is important here to highlight the temporal redundancy between the measured displacements $d_{t_0,t_6}$, $d_{t_0,t_4}$ and $d_{t_4,t_6}$. The
estimated displacement $\hat{d}_{t_0,t_4}$ can be obtained from these three displacements. This is the main interest of using temporal inversion: reducing the noise by using the redundancy between displacement measurements computed with different temporal baselines. Note that $X$ contains either the East/West displacements or the North/South displacements.

### 2.1.2 Inversion of the system

Most of the time, the system $AX = Y$ is ill-posed, *i.e.* $rank(A) < p$ (the number of linearly independent rows of the matrix $A$
is lower than the number of estimations $p$), and the system has an infinite number of solutions. To overcome this problem, the system can be solved either with a Singular Value Decomposition (SVD) (Berardino et al., 2002), a Least Square (LS) approach (Bontemps et al., 2018; Doin et al., 2011; Samsonov and d'Oreye, 2017; Charrier et al., 2022b) or a L1-norm solution (Lauknes et al., 2010). The L1-norm is more robust to outliers, but computationally expensive, as it requires computing the absolute of the residuals, which is not a differentiable piece-wise function. The SVD solution is equivalent to the minimum-norm LS solution
(*i.e.* it tends to minimize the norm of $X$) (Berardino et al., 2002). In order to have a more flexible regularization strategy, we use a Weighted Least Square (WLS) approach. The cost function is:

$$\arg\min(||W(AX - Y)||^2 + \lambda||\Gamma(X - X_0)||^2) \tag{3}$$





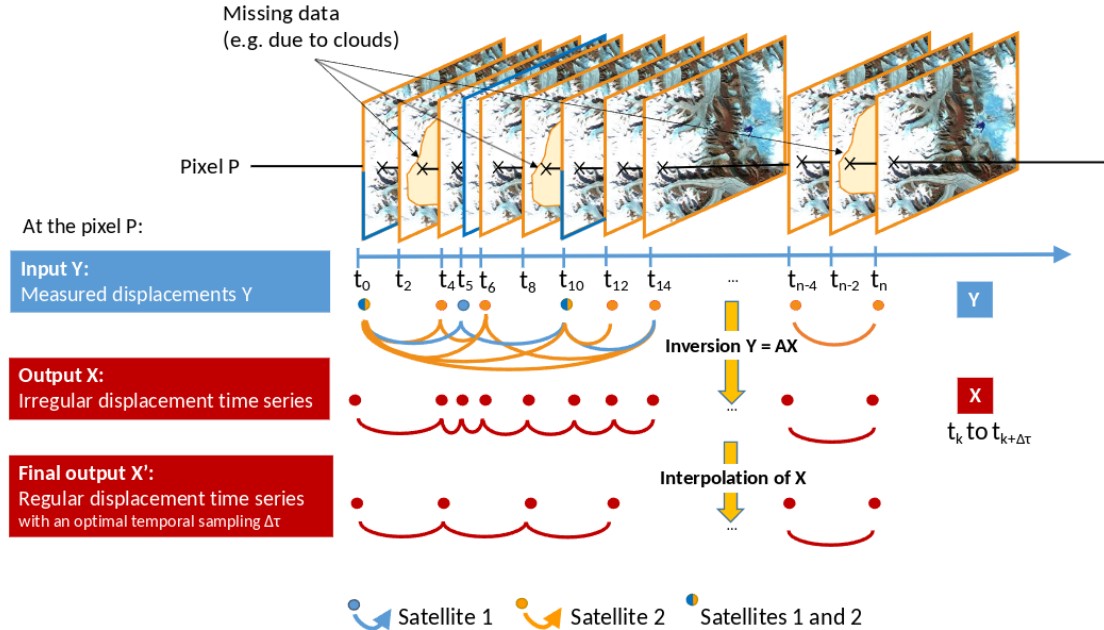

**Figure 1.** TICOI workflow. In this example, displacements have been measured using images from satellites 1 and 2, which have a repeat cycle of 2 and 5 days, respectively. Outliers have been removed in the displacement dataset, for example from areas impacted by clouds (shown in yellow on the image subsets). The TICOI workflow is applied pixelwise (in this example on the pixel $P$). First, the system $AX = Y$ is inverted to obtain an irregular time series [section 2.1.2]. Second, the time series is interpolated to obtain a regular time series with a homogeneous and optimal temporal sampling [section 2.1.3]. Note that the temporal sampling of the final time-series $\tau$ can be chosen by the end-user.

where $W$ is a $n \times p$ matrix standing for the weight given to each value in $Y$, $\lambda$ is a scaling constant and $\Gamma$ is a $p \times p$ matrix representing the regularization matrix, and $X_0$ is first guess solution detailed in section 2.2.2.

Different regularization matrix $\Gamma$ and weights $W$ can be used. The choice of $\Gamma$ will be discussed in section 2.2.2. As for the weights, $W$ could be equal to the identity (Berardino et al., 2002), in which case the WLS solution is equivalent to an Ordinary Least Square. But this ignores the heteroscedasticity of the measurements (*i.e.* the fact that they have unequal variances). Therefore, it is common to use *a priori* knowledge of the data quality, for example the InSAR coherence (Yunjun et al., 2019), the shape of the similarity map used in image correlation (Bontemps et al., 2018), errors computed over stable areas, the cosine

of the angle between each displacement vector and the spatio-temporal median, or the modified zscore (Charrier et al., 2022a). However, these metrics do not always accurately represent the errors of the measurements. Another complementary strategy is to use the inverse of the residuals (Bontemps et al., 2018; Liang et al., 2020), which can be expressed as the vector of dimension $n$:





$$R = A\hat{X} - Y \tag{4}$$

In the TICOI workflow, the weights can be a combination of *a priori* knowledge of the data quality and residuals. To be more robust to outliers, we use an Iterative Re-weighted Least Square (IRLS) approach, *i.e.* we update the weights iteratively using the residuals from the previous inversion.

In the first iteration of the inversion, if the data quality is known, the diagonal elements of the weights, $W_0$, corresponds to the errors scaled between 0 and 1.

In the second iteration, and all following iterations, each diagonal element of the weighted matrix, at the position $m, m$ and iteration $u$ is updated as:

$$W_{m,m}^u = \psi(Z_m^u, c) \tag{5}$$

In this equation, $Z$ is a standardized residual vector of dimension $n$ computed as:

$$Z = \frac{R}{\text{NMAD}(R)} \tag{6}$$

with NMAD being the Normalized Median Absolute Deviation of the residuals, equal to $1.483 MAD$.

$\psi$ is the Tukey's biweight function, which is a common down-weight function (Liang et al., 2020) robust to large outliers, defined as:

$$\psi(Z_m^u, c) = \begin{cases} [1 - (Z_m^u/c)^2]^2, |Z_m^u| < c \\ 0, |Z_m^u| > c \end{cases} \tag{7}$$

where $c$ is a tuning constant which is usually set to 4.685, producing 95% efficiency for a normal distribution (Huber, 1992).

The iterations stop when $(\text{mean}(|\hat{X}^u - \hat{X}^{u-1}|) < \delta$ or $(u > 10))$ where $\hat{X}^u$ corresponds to the results of a given iteration $u$ and $\hat{X}^{u-1}$ the results of the previous one. $\delta$ is set to 0.1 m.

### 2.1.3   Interpolation of the irregular time series

The inversion results in an irregular displacement time series, *i.e.* the vector $X$ contains displacement between each measured dates (all dates with an image acquisition minus dates rejected by outlier removal) (Fig. 1). However, irregular time series are

not always relevant (Charrier et al., 2022a). To study variations of fast moving objects such as glaciers we are interested in velocity time series (Derkacheva et al., 2020; Greene et al., 2020), more than cumulative displacement time series (Lacroix et al., 2019; Doin et al., 2011). When looking at velocities, it is necessary to compare velocities with the same temporal sampling (Charrier et al., 2022a) for three main reasons: 1) velocities with different temporal sampling are not comparable





because they correspond to the integral of the instantaneous velocity; 2) velocities with very short temporal sampling can be very noisy because noise decreases with increasing temporal sampling; and 3) the dates between which the displacements are inverted can be different from one pixel to another because of outliers removal.

To overcome this problem, some studies have proposed to use fractions of displacements inside the network (Samsonov et al., 2021). However, this assumes the velocity to be constant over the corresponding temporal sampling, which is wrong for long temporal windows due to seasonal variations or glacier surges (Charrier et al., 2022c). Therefore, we propose to: 1) compute the cumulative displacement time series by summation of the irregular time series; 2) interpolate this cumulative displacement time series (here using a cubic spline); and 3) obtain a regular time series with a given temporal sampling, using a discrete derivative. See Charrier et al. (2022a) for more details.

The resulting time series has a constant temporal sampling, for example 4 days in Figure 1. The larger the temporal sampling, the smoother the time series, *i.e.* the signal-to-noise ratio increases, but the temporal resolution decreases. By analyzing the Root Mean Square Error (RMSE) over stable areas, we have shown that the RMSE according to the temporal sampling has an asymptotic behavior which converges after around 30 days for glaciers with medium average velocity ($\sim$100 to 200 m yr$^{-1}$) (Charrier et al., 2022a, b).

## 2.2 Improved robustness and computational performance

The approach presented above performs poorly in some extreme cases, such as temporal decorrelation or abrupt non-linear changes with few image-pair velocities. Below, we show improvements to the method that overcome these challenges.

### 2.2.1 Robustness to temporal decorrelation

Using a LS approach to solve $AX = Y$ assumes the errors of $Y$ to be normally distributed. However, this assumption is not always true in a real case scenario (Fig. A1). Robust LS regression, like IRLS using Tukey's bi-weight function, helps to reduce the effect of outliers in case of random errors (Liang et al., 2020; Charrier et al., 2022b). Still, errors can also be systematic, especially when temporal decorrelation occurs: the displacements tend to be lower than expected and the distribution of errors will be heavy-tailed, with a strong kurtosis (Fig. A1). To overcome this problem, we propose to carry on a first LS with small temporal baselines only (lower than 180 days), to automatically detect temporal decorrelation. We compute the residual between each observation (with short and long temporal baselines), to this first small baseline LS solution. Displacements impacted by temporal decorrelation have large residuals, because they do not satisfy temporal closure.

### 2.2.2 New regularization term robust to abrupt non-linear changes

A crucial choice is the regularization matrix ($\Gamma$ in Eq. 3). The most common regularization matrix is the first order Tikhonov regularization, which assumes the acceleration to be null in time (Bontemps et al., 2018; Charrier et al., 2022b; Lacroix et al., 2019; Samsonov et al., 2021). For this regularization, $\Gamma$ corresponds to a first order derivative operator. The diagonal elements of $\Gamma$ in Eq. 1 are: $\Gamma_{k,k} = 1/\Delta\tau$ and the element above the diagonal will be $\Gamma_{k,k+1} = -1/\Delta\tau$ with $\Delta\tau$ the temporal sampling





of time series. When using the Tikhonov regularization, $X_0$ is a vector filled with 0 in the cost function shown in Eq. 3. In other words, this strategy minimizes the difference between adjacent estimated velocities, *i.e.* the acceleration. However, the assumption of an acceleration close to 0 is not always affordable, especially when abrupt non-linear changes occur (*e.g.*, for surge-type glaciers, or those with strong spring speedups). López-Quiroz et al. (2009) and Pepe et al. (2016) proposed to use a model-based regularization, but it also requires *a priori* knowledge of the displacement behavior.

Therefore, we propose a new regularization strategy. We constrain the acceleration of the estimated time-series, $X$, with the acceleration of an initial guess, $X_0$. In the regularization term $||\Gamma(X - X_0)||^2)$ from Eq. 3, $\Gamma$ is identical to the Tikhonov regularization matrix but $X_0$ is not null anymore. The latest is estimated from the spatio-temporal smoothing of linearly interpolated measurements, to ensure that $X_0$ and $X$ correspond to the same time intervals. We have tested different filters and have obtained the best results with a spatial window of 3x3 pixels and a third-order Savitzky–Golay filter with a temporal

window of 90 days (see appendix B1). We use only velocity measurements with temporal baselines <180 days to avoid long temporal baselines, which will be close to the long-term average of velocity, resulting in an over-smoothed solution (see Fig. A2).

### 2.2.3 Improved computational time

When velocities are derived from images acquired by different satellites, spanning different temporal baselines (*e.g.*, from 5 to

400 days in the ITS_LIVE dataset), the length of $Y$ can be very large (*e.g.*, on the order of 10,000 over Kaskawulsh Glacier, Yukon, between 2013 and 2022). This will result in very long computational time (several seconds per pixel). To mitigate this challenge, we solve the Least Square problem using LSMR, a conjugate-gradient method for sparse least-squares problems, that leverages the fact that the matrix $A$ is generally sparse, *i.e.* contains mainly 0. Additionally, we implement embarrassingly parallel processing at the pixel level. Besides, the lazy mode from Dask, an open-source Python library for parallel computing,

allows the user to directly apply TICOI on ITS_LIVE data sets stored in the Amazon cloud (López et al., 2023), without the need to download the data locally. Dask also allows out-of-memory processing by splitting the data in chunks.

### 2.3 Automatic selection of the regularization coefficient

The choice of the regularization coefficient $\lambda$ in Eq. 3 can be empirical (Bontemps et al., 2018; Lacroix et al., 2019) or based on an L-curve (Samsonov et al., 2021), which aims to compare $||W(AX - Y)||^2$ and $||\Gamma(X - X_0)||^2$ in Eq. 3. However, L-curves

do not always converge (Vogel, 1996). As an alternative solution, we propose to use the Velocity Vector Coherence (Charrier et al., 2022b; Dehecq et al., 2015) defined as:

$$\text{VVC} = \text{mean}_{(i,j) \in \omega} \left\| \sum_{t=0}^{N} \frac{\boldsymbol{V}(i,j,t)}{\|\boldsymbol{V}(i,j,t)\|} \right\| \qquad (8)$$

with $\omega$ the area over which the VVC is computed. The VVC varies from 0 to 1, with 1 corresponding to a perfect coherence of the direction in time.





When the regularization coefficient increases, the solution tends to be smoother. Therefore, the direction of the velocity vectors tends to be constant in time, and the VVC tends to be close to 1. To find a compromise between smoothing and temporal resolution, we need to find the inflection point of this function, which is approximated by:

$$VVC_{optimal\_coef} = max(VVC) - 0.05 * (max(VVC) - min(VVC)) \tag{9}$$

## 2.4 Uncertainty

Three metrics are proposed to evaluate the uncertainties of the estimated velocity time-series: 1) the VVC of the time-series; 2) the number of image-pair velocities that have contributed to each estimation; and 3) the *a posteriori* covariance matrix.

The VVC has been defined in Eq. 8. The number of image-pair velocities that have contributed to each estimation is defined as:

$$X count = A^T W \tag{10}$$

This is computed for North/South and East/West components separately. Then, the number of image-pair velocities of the velocity magnitude is taken as the average number of image-pair velocities of two velocity components.

The *a posteriori* covariance matrix is defined by assuming the errors to be independent (Gavin, 2023; Liang et al., 2020):

$$\Sigma_{\hat{X}} = N^{-1} A^T W \Sigma_{\hat{Y}} W A N^{-1} \tag{11}$$

with $N = A^T W A + \lambda \Gamma^T \Gamma$, which contains a data fidelity term $A^T W A$, and a regularization term $\lambda \Gamma^T \Gamma$. $\Sigma_{\hat{Y}}$ is the covariance matrix of the image-pair velocities, which contains the square of the errors provided with the raw image-pair velocities, converted in meters. If the errors are independent, this matrix is diagonal. The demonstration of this formula is provided in appendix A.

The *a posteriori* covariance matrix is computed for each of the components separately and interpolated using the same strategy described in 2.1.3. Then, the *a posteriori* covariance matrix of the velocity magnitude is computed following the propagation of uncertainty:

$$\Sigma_{\hat{v}} = \sqrt{(\frac{v_x}{v} * \Sigma_{\hat{v_x}})^2 + (\frac{v_y}{v} * \Sigma_{\hat{v_y}})^2} \tag{12}$$

with $v_x$ and $v_y$ the x- and y- velocity component, $v$ the velocity magnitude and $\Sigma_k$ the *a posteriori* covariance matrix of the variable k.

Finally, the confidence intervals are defined for each estimated velocity as: $\pm t_{(1-\alpha/2),n-p}\sqrt{\Sigma_{\hat{v}}}$, with $t_{(1-\alpha/2),n-p}$ the value of the student's t-distribution for a degree of freedom of $n-p$ at a confidence level of $100(1-\alpha)\%$. Here, we chose $\alpha$ to be equal to 0.05, i.e., we provide a 95% confidence interval.



## 3   Data and study sites

The package is validates on three glaciers with medium velocity magnitude ($\sim$100 to 200 m yr$^{-1}$) (Fig. 2). We chose these glaciers because they have been monitored with continuous GNSS data over multi-year periods and represent both seasonal and surge type dynamics. The possible sites available for this comparison are rare since few glaciers are monitored this way. In the following section we describe each of the glaciers, their associated GNSS data, and surface velocity measurements derived from remote sensing.

### 3.1   Lowell and Kaskawulsh glaciers in Yukon, Canada

Lowell and Kaskawulsh glaciers are two large valley glaciers located in Kluane National Park, at the eastern edge of the St Elias Mountains, in Yukon, Canada. Lowell glacier, also known as Nàlùdäy in Southern Tutchone, is a $\sim$ 65 km long surge-type glacier. It is composed of a southern and a northern arm divided by a medial moraine. The northern arm joins the main trunk by a $\sim$ 200 m high icefall, whereas the southern arm originates from a large high-accumulation basin. The terminus of Lowell is divided in two by a large nunatak (Bevington and Copland, 2014). The last 6 surges have been observed in 1948-50, 1968-70, 1983-84, 1997-98, 2009-10 (Bevington and Copland, 2014), and 2021-22 (Van Wychen et al., 2023). During the 5 surges that occurred between 1948 and 2009, the length of the surge active phase ranged from 0.6 to 2 years and the quiescent phase from 11 to 18 years. Bevington and Copland (2014) note that the surge cycle (quiescent + active phase) seems to have decreased in time, which was supported by the start of the most recent surge in 2021. Surges show a rapid terminus advance starting in summer to early fall (late June to early October), continuing through the winter, and ending in June or July of the following year. Velocity peaks during the surge phase are typically >3500 m yr$^{-1}$ in the terminus region, and the fastest-recorded motion is of 11 000 m yr$^{-1}$ in the lowest part of the south arm during the 1983–84 surge (Bevington and Copland, 2014).

Kaskawulsh, also known as Tänshį in Southern Tutchone language, is approximately 70 km long, with altitudes ranging from 800 to 2500 m above sea level (a.s.l) and flowing eastwards. It is divided into three main tributaries (North, Central, and South Arms) (Foy et al., 2011; Flowers et al., 2014). It is believed to be temperate, at least across its ablation area, and has been monitored for many decades (Clarke, 2014; Foy et al., 2011; Flowers et al., 2014; Arendt et al., 2002), starting with the Icefield Ranges Research Project in the 1960s and the 1970s (Clarke et al., 1967; Anderton, 1973). Annual velocities are about 10 to 50 m yr$^{-1}$ near the terminus, and range between 100 and 200 m yr$^{-1}$ over the rest of the centerline (Millan et al., 2022; Waechter et al., 2015; Main et al., 2023). Its surface velocity remained stable between 1960 and 2012, except for the lower part of the ablation area (up to  10 km upstream of the terminus; Waechter et al., 2015). Annual average velocities across the lower glacier increased at a rate of 5.5 m yr$^{-2}$ between 2010 and 2015, followed by a decrease between 2016 and 2018 of about 8 m yr$^{-2}$, mainly in the northern lobe (Main et al., 2023). The main cause of these changes can be explained by the drainage of proglacial Slims lake, located northwest of the terminus, according to Main et al. (2023).



## 3.2 Land-terminating margin in western Greenland

The land-terminating region in western Greenland is characterized by an absence of marine outlet glaciers where annual velocities are on the order of 100-200 m/yr (Joughin et al., 2018). The considered study area is located inland of Issunguata Sermia, a land-terminating outlet glacier, particularly well studied over the last two decades using both satellite and in situ instrumentation. Data from this region serves as the foundation for a better understanding of hydrology-dynamic coupling in Greenland, where melt-forced velocity changes have been observed from daily to decadal timescales (Davison et al., 2019). Surface lake drainage and intense melt events drive flow accelerations as high as 10 times above background across daily to weekly timescales as the subglacial drainage system is temporarily overwhelmed by the rapid influx of meltwater (Doyle et al., 2015; Tedstone et al., 2013). Seasonal velocity cycles (two-three times winter velocities) driven by summer melt production have been well-documented and attributed to basal pressure changes modulated by melt supply variability and seasonally evolving drainage (Bartholomew et al., 2012, 2010; Sole et al., 2013; van de Wal et al., 2015; Maier et al., 2022). Across decadal timescales, gradual velocity decreases (20 percent) have been found in response to periods of elevated melt rates (Tedstone et al., 2015; Halas et al., 2023; Williams et al., 2020). These variations are hypothesized to be caused by increases in summer drainage efficiency, which gradually depressurizes the ice-base (Tedstone et al., 2015; Williams et al., 2021).

## 3.3 Surface velocity measurements

### 3.3.1 Datasets

We demonstrate the TICOI method with two datasets of surface flow velocity: 1) the *Institut des Geosciences de l'Environnement* (IGE) dataset (Millan et al., 2022, 2019; Derkacheva et al., 2020; Halas et al., 2023, 2022) and 2) the NASA MEaSUREs project Inter-mission Time Series of Land Ice Velocity and Elevation (ITS_LIVE) dataset (Gardner et al., 2018, 2022). The first one derived displacement using a modified version of the Normalized Cross-Correlation (NCC) algorithm AMPCOR from ROI_PAC (Millan et al., 2019, 2022). In the Yukon, it is based on images from Sentinel-2 and Landsat-8 and the correlation window size is 16x16 pixels. In Greenland, the image-pair velocities are based on Landsat-7, Sentinel-2, Landsat-8 and Sentinel-1 data and corresponds to datasets published by Halas et al. (2023); Derkacheva et al. (2020). Image-pair velocities span temporal baselines from 5 to 100 days and from 330 to 400 days in Yukon, from 5 to 32 days (Derkacheva et al., 2020) and from 330 to 400 days over western Greenland (Halas et al., 2023). The spatial sampling of the velocity maps is 50 m in Yukon and 150 m in Greenland. In the dataset, a previous outlier filtering has been performed: displacements that deviate more than three pixels from the median velocity computed over a spatial window of $9 \times 9$ pixels are considered to be outliers (Millan et al., 2019; Mouginot et al., 2012). The uncertainties depend on the spatial resolution of the images and the temporal baselines as explained in Millan et al. (2019), by assuming the uncertainty in displacement to be around 1/10 pixels.

The second dataset, ITS_LIVE v2.0 (Gardner et al., 2022), contains velocities measured using the NCC algorithm "autoRIFT" (autonomous repeat image feature tracking) on images from Sentinel-1, Sentinel-2, Landsat-7 and 8. Results from the sparse search guide a dense search (Gardner et al., 2018). The size of the correlation window is increased iteratively according to the normalized displacement coherence, an indicator of the quality of the correlation (Gardner et al., 2018). The temporal





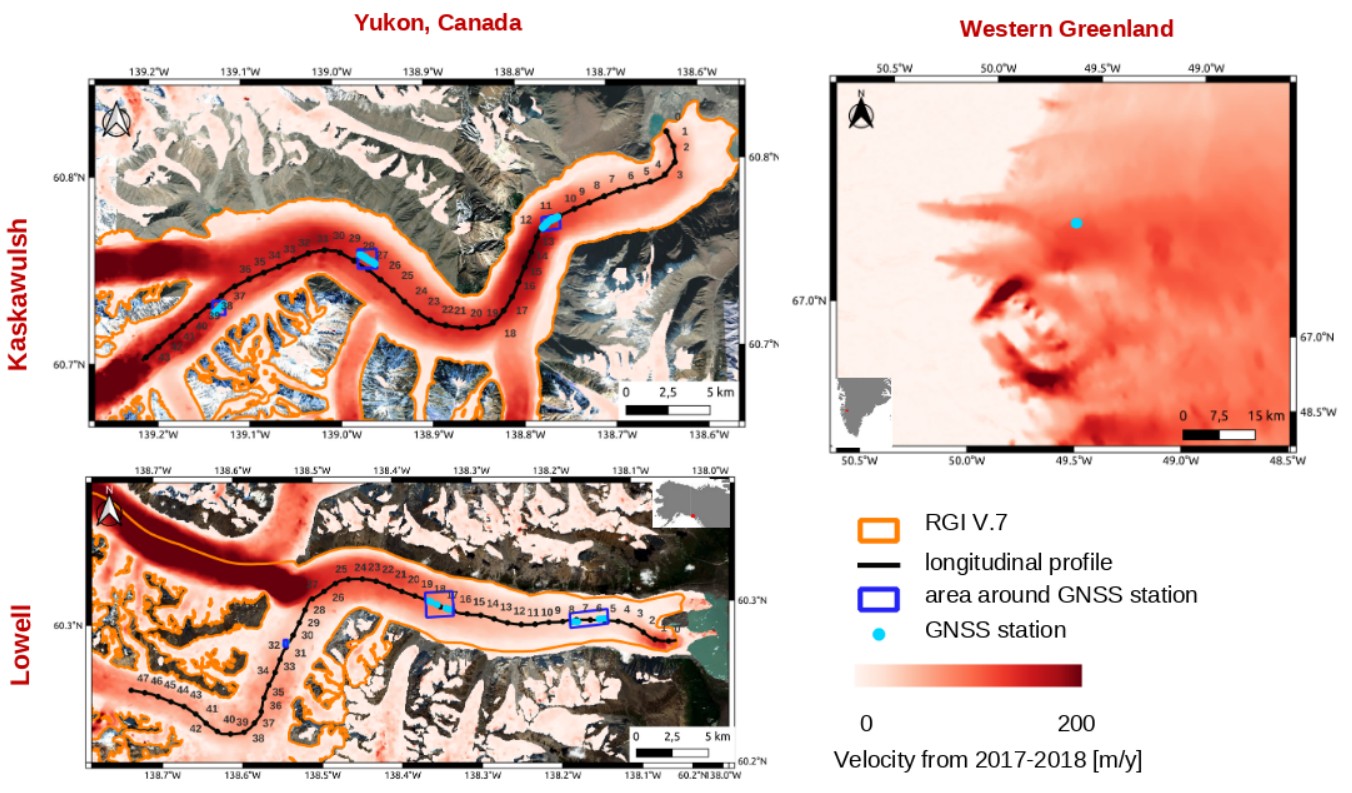

**Figure 2.** Study areas of three selected glaciers: Kaskawulsh and Lowell glaciers in the Yukon; Issunguata Sermia Glacier in western Greenland. Black lines represent the center flow-line of Kaskawulsh and Lowell glaciers according to the Randolph Glacier Inventory version 7 (RGI 7.0 Consortium, 2023). It has been slightly modified for Kaskawulsh to reach Slims lake. Orange lines represent the glacier outlines from RGI v7. Blue dots show time evolving locations of GNSS stations. The three main GNSS locations are named Kask L, M, U for the lower, middle and the upper of Kaskawulsh glacier, and Lowell L, M, U for the lower, middle and upper of Lowell glacier. Blue squares represent areas including all GNSS stations for each site, they are used in section B. The background of each site is the 2017-2018 average velocity from Millan et al. (2022) overlaid over Sentinel-2 images.

baselines range between 5 and 546 days. The spatial sampling of the velocity maps is 240 m resampled at a resolution of 120 m using a cubic spline approach (Lei et al., 2022). Only velocities that agree within 4 times the centered $5 \times 5$ mean absolute deviation are retained. The uncertainty of each image-pair velocity corresponds to the standard error of velocities relative to the stable or slow moving areas.

### 3.3.2   Pre-processing of datasets

In the TICOI method, we optionally apply a filtering to the input data. Two main types of strategies for deleting outliers are implemented : the Modified Zcore (Mzscore), which filters the velocity components 3.5 NMAD away from the median of the





entire time period (Maronna et al., 2019), and the Median Angle (MA) that removes image-pair velocities that deviate more than 45° from the median vector (Rabatel et al., 2023; Charrier et al., 2022a). In this study, we use the MA filter.

The IGE and ITS_LIVE datasets are reprojected using a nearest-neighbor interpolation on the same coordinate system: the polar stereograpghic North projection (EPSG code 3413) for both Greenland and Yukon. This implies reprojecting both the coordinates of the dataset and each of the velocity components. To reproject velocity components, we first compute the coordinates of the start and end point of each vector in the new coordinate system, then we compute the difference between these two coordinates along the axis of the new coordinate system.

## 3.4 GNSS data

On Kaskawulsh and Lowell glaciers (Yukon), up to six dual frequency GNSS receivers recorded their position at 15 s intervals for 2 or 3 hours per day in winter, and 24 hours per day in summer (Waechter et al., 2015; Van Wychen et al., 2023), from 2013 to 2022. The positions were post-processed using Natural Resources Canada's Precise Point Positioning service (http://webapp.geod.nrcan.gc.ca/geod/tools-outils/ppp.php ), resulting in an accuracy of ∼1–2 cm in horizontal and ∼5 cm in vertical. We removed any position derived from a daily record of <1.5 hours, which sometimes occurs in winter due to low battery power. Prior to 2017, the stations were Trimble R7 units mounted on poles drilled into the glacier surface. From 2017 onwards, the stations were replaced with Trimble NetR9 units mounted on tripods 'floating' on the glacier surface. Every few years they were manually moved up glacier to compensate for the glacier displacement, to ensure that they remained in approximately the same location. This creates artifacts in the position time-series, which we remove using the Local Outlier Filter (LOF) (Breunig et al., 2000), which computes the local density deviation of a given data point with respect to its neighbors. We apply it on the gradient of the East/West and North/South position. A daily position is considered to be an artifact if LOF > 5. Then, velocities are derived from the discrete derivative of the position time-series, and averaged using a temporal window of 5 days, which corresponds to the minimal repeat cycle of the satellites used. Time spans with less than 80% available daily velocities are removed.

In western Greenland, we derived velocities from 15 s position data collected at a field site located 33 km from the terminus of Issunguata Sermia using five Trimble NetR9 GNSS receivers mounted on a poles frozen into the ice. The position data from each receiver was processed against an off-ice base station using TRACK v1.29 differential kinematic processing software (previously detailed in (Maier et al., 2019, 2022)). From the GPS positions, ice velocity is estimated on a daily basis (Halas et al., 2022). While the collection period was from 2014-2017, no data are available from each winter due to power limitations.

Since the Yukon GNSS stations typically move around 100-150 m yr$^{-1}$, we compare them to remote sensing velocities or TICOI results located at the averaged GNSS location of the corresponding year. The GNSS station in Greenland is supposed to be stable spatially because the period of measurements is shorter and the sampling of the velocity maps coarser (considering a spatial sampling of 150 m, an average velocity of 125 yr$^{-1}$, the GNSS stations have likely moved by 2.5 pixels between 2014-17). Then, we average daily GNSS velocities to match the same temporal baselines as remote sensing velocities, or the temporal sampling of TICOI time-series, which in this study is 30 days.





## 4 Results

### 4.1 Robustness of TICOI method

#### 4.1.1 Robustness to temporal decorrelation

One of the main improvements of TICOI compared to previous methods is its ability to eliminate artifacts caused by temporal decorrelation in the input datasets. The strategy detailed in section 2.2.1 rejects biased long-temporal-baseline velocities. These velocities tend to have large errors when compared to the temporal closure solution derived using only short-baseline velocities. This results in large residuals (Eq. 4) relative to the initial solution with small temporal baselines, and as a result, Tukey's biweight function assigns a weight of 0 to these raw image-pair velocities.

We applied TICOI to a large area around Kaskawulsh Glacier for the period 2013-22 considering two implementations: with and without automatic detection of temporal decorrelation. Then, we computed the averaged velocity magnitude of TICOI results obtained with the two implementations. The median difference between the two is 0.0 m yr$^{-1}$ (Fig. 3b). However, the difference can reach up to 100 m yr$^{-1}$ near glacier margins or in narrow, steep areas, where temporal decorrelation is more likely to occur (Fig. 3a), as in point A (Fig. 3c). Over stable areas, the difference has median values of 0.0 m yr$^{-1}$ and a MAD of 0.8 m yr$^{-1}$.

Notably, TICOI remains robust to temporal decorrelation without this specific strategy, when decorrelated image-pairs are in the minority (Fig. 8). In such cases, the Tukey biweight function is able to effectively filter out the decorrelated image-pairs.

#### 4.1.2 Robustness to strong changes in velocity

The traditional regularization penalizes abrupt changes in velocity, and is thus unable to resolve the peak of velocity for a surge event. For instance, for the surge of the lower station of Lowell glacier, the traditional regularization retrieves a continuous increase in velocity from mid-2021 onward, while TICOI captures well a sudden increase in velocity and the rapid glacier slow-down in summer 2022 (Fig. 4). In this example, few velocity measurements are available in 2022-23, because the outlier removal step of ITS_LIVE rejected many image-pair velocities (see section 3.3), therefore the solution is strongly impacted by the regularization term, which minimizes the acceleration in the traditional approach. The TICOI regularization (described in section 2.2.2) relaxes this assumption of minimal acceleration by using an initial guess about the acceleration, which makes it possible to capture the peak of the surge even with limited image-pair velocities (Fig. 4).

### 4.2 Validation with GNSS time-series for different glacier dynamics

We evaluate TICOI time-series with seven GNSS time-series on three different glaciers (Table 1). The metrics for evaluation are the RMSE (in m yr$^{-1}$) and Kling–Gupta efficiency (KGE) (no unit). The KGE is a goodness-of-fit indicator, widely used to calibrate hydrological models, in order to make sure that they well capture peak flows, as well as the seasonality of the flow. It is defined as:





**Figure 3.** Illustration of the robustness to temporal decorrelation over Kaskawulsh glacier: a) represents the difference between TICOI with and without an automatic detection of temporal decorrelation. White lines correspond to glacier outlines according to RGI v.7 (RGI 7.0 Consortium, 2023). b) illustrates the histogram of this difference over the area. c) provides an example of the time-series of ITS_LIVE image-pairs (yellow), TICOI (purple) and TICOI with an automatic detection of temporal decorrelation ("TICOI_detect_temp", red). The time-series are located at point A represented in a).

$KGE = 1 - \sqrt{(r-1)^2 + (\alpha-1)^2 + (\beta-1)^2}$  (13)




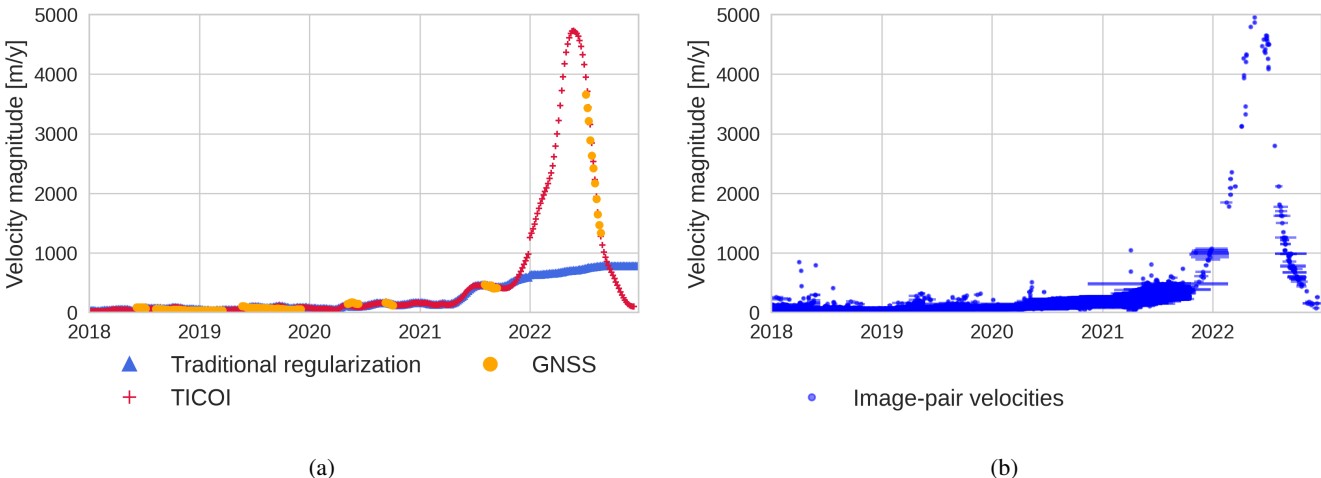

**Figure 4.** (a) Example of velocity time series retrieved on the lower part of Lowell glacier (Lowell L GNSS station) with TICOI regularization term based on an initial guess about the acceleration (red crosses) and with the traditional regularization term (blue triangles), *i.e.*, the first order Tikhonov regularization. The orange dots show the GNSS data averaged to match TICOI temporal sampling (note data gaps during the first part of the surge). (b) Image-pair velocities from ITS_LIVE at Lowell L. Dots and bars represent the central date and temporal baseline, respectively, of each image-pair velocity.

with $r$ the Pearson's coefficient, $\alpha = \frac{\sigma_e}{\sigma_o}$ with $\sigma_e$ and $\sigma_o$ the variance of the estimated and observed time-series, respectively, and $\beta = \frac{\mu_e}{\mu_o}$ with $\mu_e$ and $\mu_o$ the mean of the estimated and observed time-series, respectively. $\alpha$ represents the variability of the estimation and $\beta$ is the bias term. A perfect agreement between two time-series would lead to a KGE of 1.

TICOI leads to a reduction in RMSE from 9 to 69% in comparison with the image-pair velocities, with a median improvement of 52%, and an increase in KGE from -16 to 87%, with a median improvement of 62% (Table 1). This improvement is less significant for the Lowell L and M stations, due to a larger density of TICOI estimation during the surge in comparison with the image-pair velocities. Besides, TICOI leads to a reduction of RMSE from 11 to 81% in comparison with a rolling median, with a median improvement of 40%, and an increase in KGE from 21 to 84%, with a median improvement of 55%. For the point Lowell U, the rolling median provides better results than TICOI, this could be due to the low signal-to-noise ratio

of image-pair velocities over this area. A strategy to improve this result is discussed in section 5.2.

We note that a rolling median using all temporal baselines gives slightly better RMSE than a rolling median with small baselines only, for some of the considered points, as illustrated in Figure A2. However, the solution using all baselines tends to underestimate the largest velocities, because they include annual temporal baselines which are close to the annual average of the signal. These results show the strength of TICOI: taking advantage of all temporal baselines, without over-smoothing the

solution.





| RMSE | TICOI | Image-pairs | Moving median |
|---|---|---|---|
| Kask L | **21.3** | 68.4 | 28.3 |
| Kask M | **28.4** | 62.5 | 31.8 |
| Kask U | **43.1** | 90.5 | 55.5 |
| Lowell L | **58.8** | 64.5 | 308.7 |
| Lowell M | **44.7** | 60.2 | 155.7 |
| Lowell U | 78.7 | 119.9 | **53.7** |
| Iss | **12.5** | 28.8 | 28.2 |

| KGE | TICOI | Image-pairs | Moving median |
|---|---|---|---|
| Kask L | **0.84** | -0.27 | 0.71 |
| Kask M | **0.78** | -0.14 | 0.72 |
| Kask U | **0.08** | -1.79 | -0.36 |
| Lowell L | **0.92** | 0.86 | 0.5 |
| Lowell M | 0.93 | **0.94** | 0.66 |
| Lowell U | 0.49 | 0.53 | **0.74** |
| Iss | **0.86** | 0.63 | 0.59 |

**Table 1.** Comparison of the RMSE (in m yr$^{-1}$, left) and KGE (unitless, right) of the velocity magnitude between GNSS and: 1) TICOI time-series; 2) image-pairs with a temporal baseline lower than 180 days, and 3) a moving 30 days median applied on image-pairs with a temporal baseline lower than 180 days. The lowest values are in bold. The surface velocity measurements used are the ITS_LIVE data in the Yukon, and IGE data in Greenland.

## 4.3 Sensitivity analysis and automated choice of the hyperparameters

TICOI was developed as a flexible method, with processing options that can be changed by the user. Several options can be modified: the coefficient of regularization, the possibility to set an initial weight, the strategy to delete outliers, the type of spatio-temporal filter, and the solver.

The regularization coefficient has the greatest impact on the TICOI solutions. To illustrate this, we compute the median RMSE and KGE for the velocity magnitude at the six Yukon GNSS stations (Fig. 5 a). Both RMSE and KGE improve significantly, by factors of 3 and 8 respectively, when the regularization coefficient increases from 0.1 to the optimal value of 100. This optimal coefficient corresponds to $1.1 \min(RMSE)$ and $0.9 \max(KGE)$.

     Notably, the RMSE increases slightly and the KGE decreases slightly (by about 5%) when the coefficient is further increased 390 to 10,000, indicating the relative stability of the solution even with large regularization coefficients in general scenarios. However, for surge-type glaciers, using a coefficient greater than 1000 increases the risk of over-smoothing the solution (Fig. B1).

     The comparison between TICOI results and GNSS data helps to identify an optimal regularization coefficient of 100. In many cases, though, GNSS observations are unavailable to optimize the regularization coefficient. In such cases, we suggest that the optimal coefficient can also be determined as the approximate point of inflection of the VVC curve, defined in Eq. 9. 395 This method similarly yields an optimal value of 100 for the coefficient. Thus, the VVC approach provides a reliable method for selecting an optimal regularization coefficient (Fig. 5 b).

     We analyzed the sensitivity of the results to various options in appendix B. Our analysis shows that the choice of weight and solver has no impact on the results for the tested cases. The selection of spatio-temporal filters introduces a standard deviation in RMSE of approximately 2.6 m yr$^{-1}$ on average between filters (*i.e.* 8% of the averaged RMSE). For non-surge glaciers,





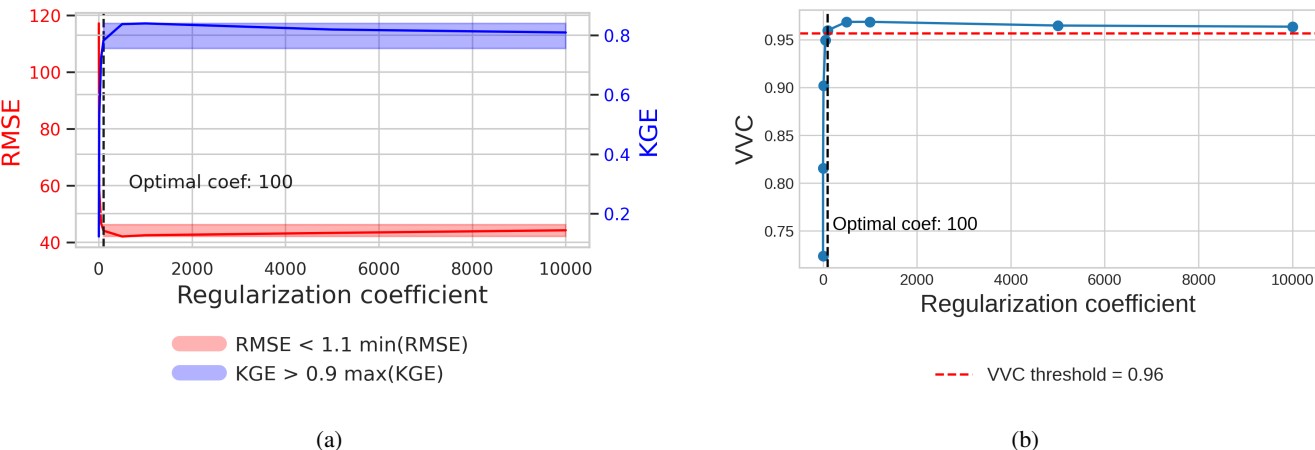

(a)                                                          (b)

**Figure 5.** (a) Evolution of the RMSE and KGE of TICOI results with respect to the six Yukon GNSS stations according to the regularization coefficient. (b) Evolution of the VVC computed over the six red squares shown in Figure 2 as a function of the regularization coefficient. The optimal coefficient corresponds to $max(VVC) - 0.05 * (max(VVC) - min(VVC))$. The optimal coefficient found is 100 using both approaches (a) and (b), which confirms the value of using VVC for selecting the regularization coefficient.

using MA filters increases the RMSE by up to 2 m yr$^{-1}$ compared to a strategy that does not filter outliers. This is largely because the Tukey biweight function, which is applied, already acts as an outlier filter.

### 4.4 Uncertainty of the final product

To provide insights into the uncertainty of TICOI velocity time-series, we first use the VVC per pixel. This highlights areas where the direction of TICOI results has a poor temporal coherence. In Figure 6 we can see that the VVC is lower near 405   the terminus, the borders and the upper parts of the glaciers. We also notice low VVC values (around 0.7) just before the confluence between the southern arm and the main trunk of Lowell glacier. This corresponds to the approximate position of the equilibrium-line altitude, therefore leading to strong changes in surface state (between wet snow, dry snow and bare ice) (NASA Earth Observatory, 2018) that are difficult to tackle. It is also the only part of the glacier showing a northern aspect, so is more likely to be impacted by shadows and illumination changes (Lacroix et al., 2019).

Then, to estimate the uncertainty of each TICOI retrieved velocity, in time and space, we propagate the covariance matrix as described in section 2.4. We tested this approach on simulated data with correlated noise described in appendix C, for different percentage of image-pair velocities and different noise levels. In controlled conditions, the 95% confidence interval includes both estimated and true velocity for more than 95% of the estimation, except for low percentage of data and low noise where the confidence tends to be slightly underestimated (Fig. 7). This synthetic case provides confidence of the relevance of the 415   theoretical framework.

     In order to test the relevance of the uncertainty calculation, we calculate the 95 % confidence intervals for the real dataset of Kask L (Fig. 8). Confidence intervals are larger before the launch of Sentinel-2 in 2015/2016, when the number and quality



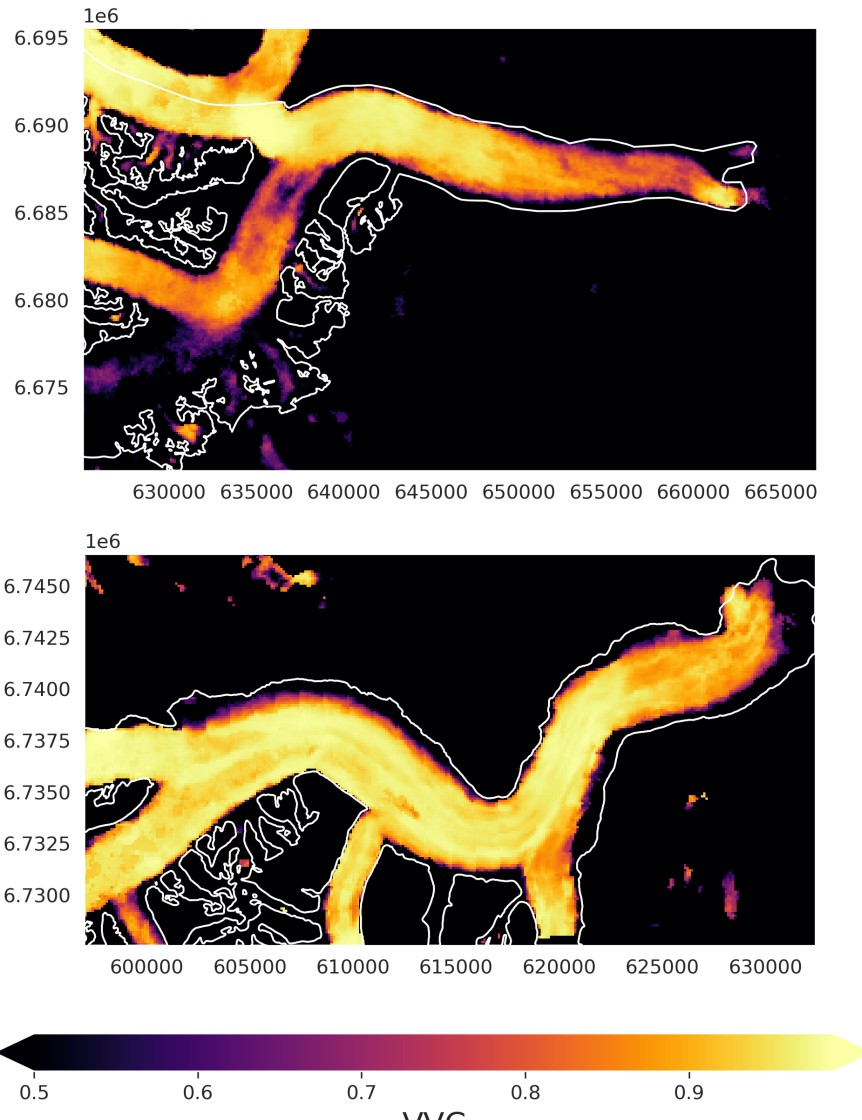

**Figure 6.** Spatial variability of VVC, overlaid with the glacier outlines in white according to the RGI v.7 (RGI 7.0 Consortium, 2023), over the same areas as in Figure 2. The top and bottom panels represent Lowell and Kaskawulsh glaciers, respectively. The coordinates correspond to the projection EPSG:32607.

of data were lower. They are also larger in wintertime for the same reason. However, only 48% of the confidence intervals include both the estimated and the GNSS velocities, which is much below the expected 95%. On average, over the six GNSS stations, the percentage is 27%. The confidence intervals fail to include estimated and true velocity, especially when the number of image-pair velocities is low, and the solution less strongly constrained. This is a limitation of our approach to calculate confidence intervals that is discussed in section 5.2.



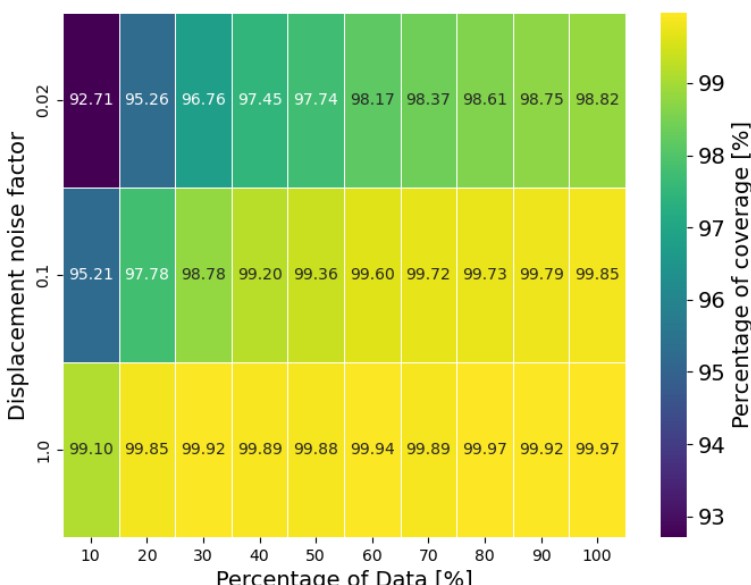

**Figure 7.** Percentage of the estimated 95% confidence intervals that include both the estimated and the true displacement values, using simulated data described in appendix C. The displacement noise factors correspond to a factor of the simulated instantaneous velocity amplitude. A factor of 1, 0.1 and 0.02 are equivalent to a standard deviation of 4.6 m, 0.4 m and 0.09 m, respectively. The percentage of data corresponds to the percentage of simulated image-pair displacements in comparison with the total number of possible image-pair displacements. The value of each configuration of displacement noise factor and percentage of image-pair displacements relies on 50 experiments.

## 4.5 Application to different glacier dynamics

In this section, we apply the TICOI method to pixels sampled regularly along the centerlines of Lowell and Kaskawulsh glaciers

(Fig. 9). On Lowell glacier, we observe an upward propagation of the surge in 2021-2022 (Fig. 9a). This surge was first reported by Van Wychen et al. (2023) using Radarsat Constellation Mission data, but since that data was only available from winter 2022 the start of the surge was poorly defined. From TICOI results, it is clear that there is a strong positive velocity anomaly in June 2021, with maximum anomalies of 400 m yr$^{-1}$ at around 2 km from the terminus (Fig. 9a). Velocity anomalies range between 100 and 400 m yr$^{-1}$ from 1 to 7 km upglacier from the terminus. They remain positive up to 27 km from the terminus, at a

location which corresponds to the confluence between the northern and southern arm of Lowell. This positive anomaly remains stable until December 2021 (with an average monthly acceleration of 15 m yr$^{-2}$) except in the 2 last kilometers before the terminus, where the average acceleration was of 1500 m yr$^{-2}$. The entire glacier starts to accelerate in December 2021, with an acceleration front propagating upglacier. Anomalies > 300 m yr$^{-1}$ are recorded up to 27 km from the terminus, just one month later, in January 2022. However, this acceleration front seems to propagate at a slower rate in the southern arm of Lowell

glacier, even if this observation has to be interpreted cautiously regarding the low VVC values obtained in this area (Fig. 6).



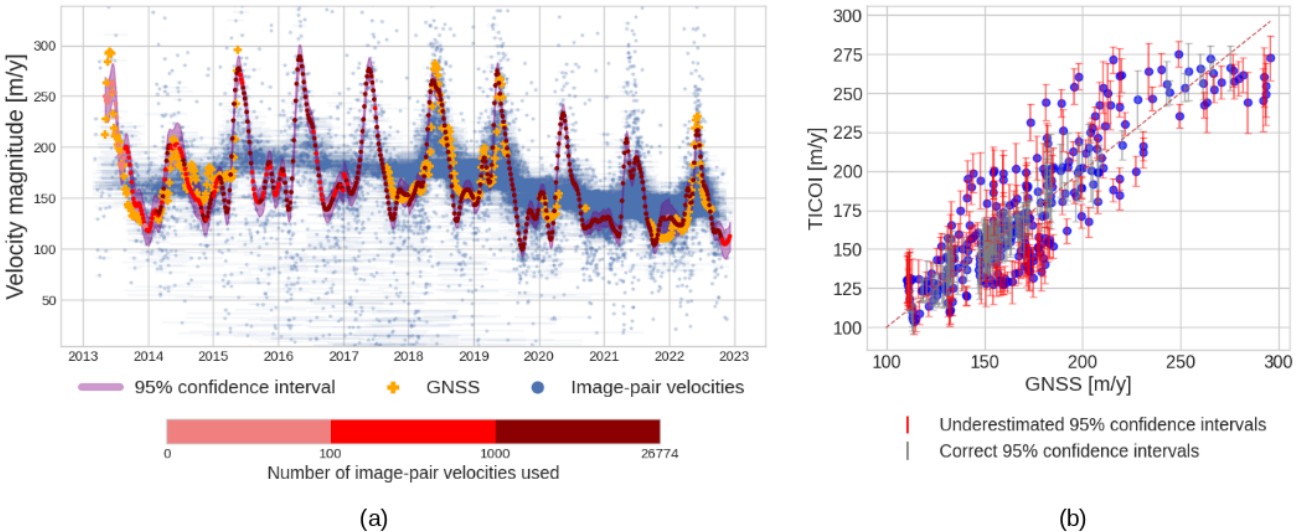

(a)                                                           (b)

**Figure 8.** (a) 95% confidence interval for velocities on the point Kask L. The blue dots and bar correspond to the central date and temporal baselines of the image-pair velocities. GNSS data are represented by orange crosses. Light magenta intervals represent the confidence intervals defined in 2.4. The colors from light red to dark red correspond to the number of image-pair velocities used to constrain the velocity estimations. The upper and lower limits of the y-axis are defined according to the average $\pm$ the standard deviation of the image-pair velocities. (b) TICOI velocity magnitude as a function of the GNSS velocity magnitude. Vertical grey bars correspond to the confidence intervals, which should intersect the red line 1:1 if they are not underestimated. Underestimated confidence intervals are displayed in red, correct one are represented in grey.

In October 2022, anomalies $> 300$ m yr$^{-1}$ are recorded up to at least 47 km from the terminus. In addition, we observe slight positive anomalies occurring each spring before the surge. The intensity of these anomalies increases from 2016 until the start of the surge, especially near the terminus, with annual maximum velocity magnitudes raising from 90 m yr$^{-1}$ between 2014-16 to 275 m yr$^{-1}$ in 2020, at a rate of 9, 33, 70 and 70 m yr$^{-2}$ in 2017, 2018, 2019 and 2020 (Fig. 10a). The annual maximum
over the entire glacier rises later in time (Fig. 10a). The surge event ends near the terminus in winter 2022, while anomalies remain high in the upper part of the glacier.

On Kaskawulsh glacier, there is a long-term trend towards decreasing velocities (Fig. 9b), particularly over the lower part of the ablation area, which is likely related to the thinning of the glacier (Main et al., 2023; Dehecq et al., 2019). In the 0 to 5 km section upglacier from the terminus this decrease in velocity between 2015 and 2018 is particularly marked, from 170-180 m
yr$^{-1}$ to 120 m yr$^{-1}$ (Fig. 10b). This is consistent with the results of Main et al. (2023), who found a significant change in velocity near the terminus after drainage of the adjacent proglacial Slims Lake in 2016. We also observe a marked seasonal velocity increase every spring (around April), with maximum of yearly anomalies between 30 and 70 m yr$^{-1}$. The anomalies propagate up glacier, but the interpretation is more difficult above km 30 from the terminus because the signal becomes noisier





(Fig. 9b). The high temporal resolution also reveals very abrupt positive anomalies in March at a position situated from 6 to 9 km from the terminus. This is especially the case for the year 2019, but it can be also noticed in 2017, 2020, 2021, and 2022. This phenomenon is also visible in the GNSS time series (Fig. 8).

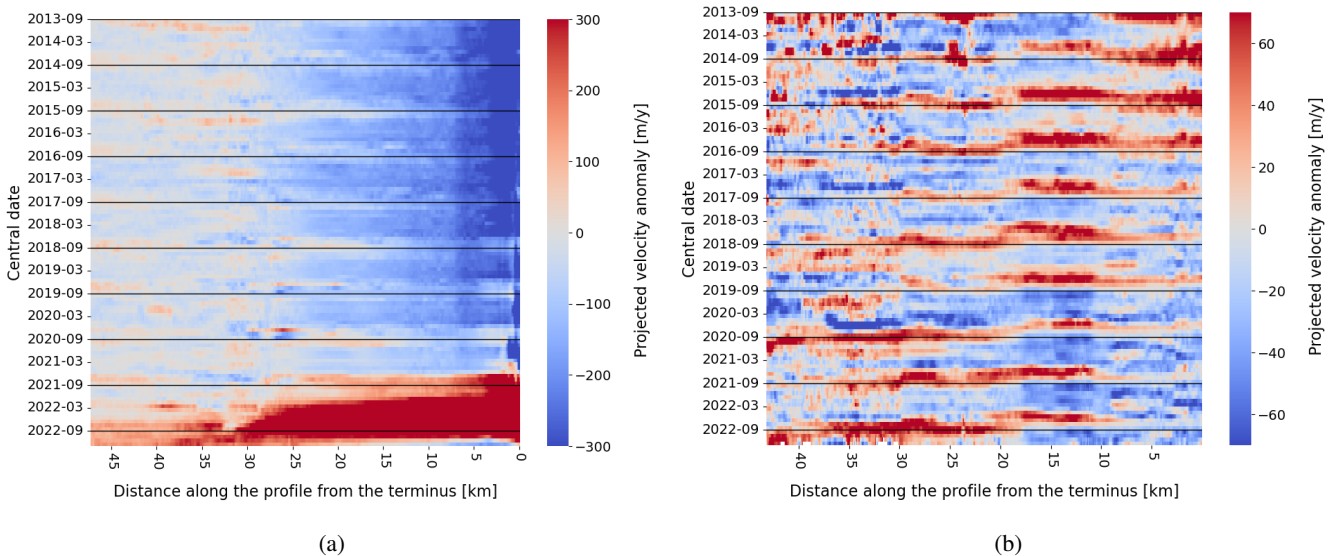

(a)                                                                                          (b)

**Figure 9.** Spatio-temporal evolution of monthly velocity anomalies with respect to the averaged velocity magnitude over the period (as defined in Dehecq et al., 2019) over the centerline of: (a) Lowell glacier, and (b) Kaskawulsh glacier, plotted as distance from the terminus. The centerline is represented in black on Figure 2. Note the logarithm scale in (a).

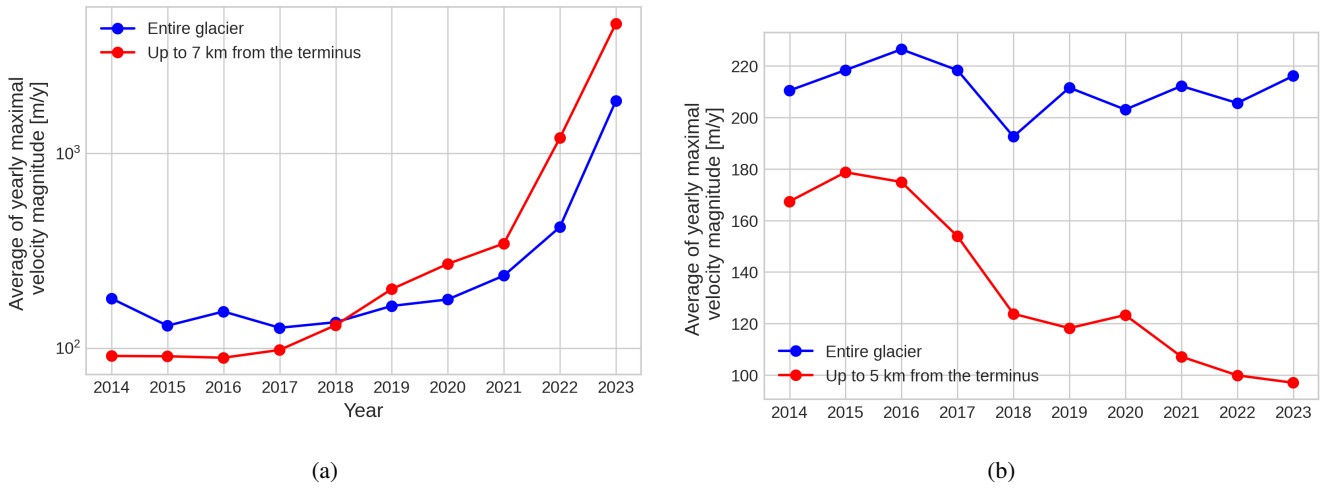

(a)                                                                                          (b)

**Figure 10.** Average yearly maximal velocity magnitude over the entire glacier in blue, and near the terminus in red, for: (a) Lowell glacier, (b) Kaskawulsh glacier.



### 4.6 Estimating monthly velocities from annual velocities

The TICOI method can be used to study sub-annual velocities even if long temporal baselines of velocity only are available. This is important because over slow moving areas, velocities quantified with small temporal baseline mainly show low signal-to-noise ratio, and are frequently filtered out in open-source datasets such as ITS_LIVE (Gardner et al., 2018; Millan et al., 2019). For those reasons, previous research chose to perform image correlation on images spaces by long time intervals only, for example between 330 and 400 days (Halas et al., 2023), privileging accuracy over temporal resolution. This strategy is adapted when assessing multi-annual or decadal velocity trends, but hinders the seasonal variations and rapid summer velocity changes (Fig. 11). In the case of Issunguata Sermia glacier, the satellite observations show an average velocity of $\sim$125 m yr$^{-1}$ with slight variations from year to year (Fig. 11). The application of TICOI to this dataset provides monthly velocities which match very well with the GNSS data after 2016, when Sentinel-2 images become available (Sentinel-2A was launched in June 2015), resulting in an increasing number of image-pair velocities (red and dark red in Figure 11). The RMSE between GNSS and TICOI time series is 25.8 m yr$^{-1}$ between 2014 and 2017, and 17.7 m yr$^{-1}$ between 2016 and 2017, with stronger errors in wintertime when optical images are impacted by night and clouds. Hence, TICOI can retrieve monthly velocities using only image-pair velocities with long temporal baselines. It takes advantage of the temporal closure which relies on redundancy of annual velocities, having Sentinel-2 providing new images every 5 days in optimal conditions. However, it still requires a sufficient amount of observation to obtain a reliable time-series (> 500).

## 5 Discussion

### 5.1 Fusion of velocity measurements from different processing chains

Datasets from several processing chains can be included as input in TICOI, with the datasets reprojected on the same grid as explained in section 3.3.2. Fusing different datasets, like the ones from IGE and ITS_LIVE, can particularly improve the signal-to-noise ratio in the upper parts of glaciers (Tab. 2). For example, this provides an improvement in RMSE of 11% for the upper part of Lowell glacier, and 32% for the upper part of Kaskawulsh glacier. Over these two areas the surface is snow-covered most of the year, which produces poor results from image correlation algorithms. Fusing velocity results from different processing chains takes advantage of the different sets of correlation parameters used and strategies to delete outliers. However, we do not see improvements in the middle and lower parts of the glaciers when applying this technique, where the RMSE even slightly decreases (Tab. 2). Considering the increasing computational time when including additional dataset, this technique may only be suited for very noisy areas or areas with a low number of image-pair velocities.

### 5.2 Uncertainty

Despite providing satisfying confidence intervals on synthetic data, our framework to calculate TICOI confidence intervals underestimates uncertainties on real data when compared to GNSS measurements (see section 4.4). Here, we discuss the different limitations in our approach that could explain these discrepancies. First, the error of the input image-pair velocities





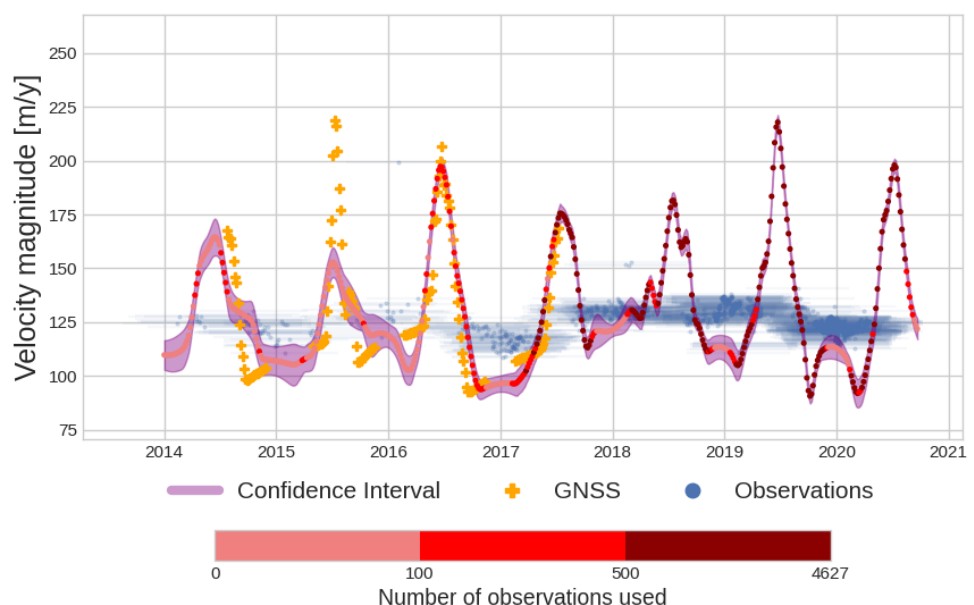

**Figure 11.** Example of monthly velocity time series (in red) retrieved using annual velocities from Halas et al. (2023) (in blue) on the Issunguata Sermia glacier. GNSS data are represented by orange crosses. The colors from light red to dark red correspond to the number of image-pair velocities used to constrain the velocity estimations.

|  | ITS_LIVE | IGE & ITS_LIVE |
|---|---|---|
| **Lowell U** | 46.2 | **41.5** |
| **Kask U** | 46.7 | **35.3** |
| **Kask M** | **26.0** | 28.5 |
| **Lowell M** | **30.1** | 33.7 |

**Table 2.** Comparison of the RMSE of the velocity magnitude (in m yr$^{-1}$) between GNSS and TICOI time-series, with datasets from 1) ITS_LIVE and 2) IGE & ITS_LIVE over upper GNSS stations of Lowell and Kaskawulsh glaciers. We consider the time-period which is common between the two datasets, i.e. 2016-22. The lowest values are in bold.

may be underestimated. These errors are computed either using stable areas, which may not be representative of the glacier texture (Altena et al., 2019, 2022) or by assuming a 1/10 pixel error, which corresponds to an error of 10 m for Sentinel-2 and
30 m for Landsat-8. To estimate the underestimation of the errors provided with image-pair velocities, we consider the true error as the difference between ITS_LIVE and GNSS data. The averaged RMSE between the errors provided by the ITS_LIVE dataset and the true errors are 9 m and 7 m for the East/West and North/South components, respectively. The difference



between the average true error and ITS_LIVE error is of 1.8 and 2.6 m, highlighting a clear underestimation in ITS_LIVE errors. Moreover, there are no strong temporal correlations between the provided errors and the true errors, with Pearson's coefficient of 0.27 on average for both components. Second, the underestimation of our confidence intervals could be caused by biases in the image-pair velocities, for example due to seasonal illumination changes (Lacroix et al., 2019). Our confidence interval only account for random errors and not systematic biases. Third, the real errors may have a stronger correlation in time than what has been simulated, for instance due to seasonal source of errors (shadows, surface changes), since we only take into account the correlation of errors between displacement with common acquisition dates. In case of highly correlated errors, the *a priori* covariance matrix cannot be diagonal anymore, leading to much higher computational cost. The cost of explicitly computing the inverse of the error covariance matrix is proportional to $n$ (the number of image-pair velocities) if the matrix is diagonal and to $n^3$ in the general case (Ruggiero et al., 2016).

To address this issue, a potential solution involves scaling the confidence interval by a specific correction factor. This factor could be a function of the VVC, as the VVC serves as an effective proxy for the relative uncertainty between pixels. Indeed, the RMSE decreases linearly with the VVC, until reaching a vertical asymptote close to 1, which is the VVC maximum value (Fig. 12a). To estimate this correction factor, we determine the 95th quantile of the theoretical 95% confidence intervals divided by the true errors, for each GNSS stations (Fig. 12b). We then perform a linear regression on the resulting seven points. Although the RMSE of the linear regression is around 3 and the sample size is small, this method provides an empirical correction factor. By multiplying the confidence intervals by this factor, the confidence intervals contain both estimated and GNSS data with an average percentage of 86% (against 27% without a correction factor). Additionally, the reliability of the confidence intervals derived with the correction factor appears to improve with the number of observations used. For example, when selecting only TICOI estimations with more than 500 observations, the confidence intervals encompass both estimated and GNSS data with an average coverage of 91%.

Another strategy could be to augment the observation vector with the first- and second-order spatial derivatives of the original observations, as described in Ruggiero et al. (2016); Brankart et al. (2009). However, this requires proper characterization of the spatial and temporal correlation of errors of surface velocities, which could be the scope of future research. With the current state of knowledge in velocity errors, we recommend relying on the VVC and number of contributed image-pair velocities.

### 5.3 Large scale application

Here, we discuss the possibility of applying TICOI at a large scale. We have shown that it can be applied to all kinds of glacier dynamics because it does not include any *a priori* information about the glacier behavior, unlike a wide range of post-processing approaches (Greene et al., 2020; Riel et al., 2021; López-Quiroz et al., 2009; Samsonov et al., 2021). This flexibility also allows for the detection of unexpected events and trends, such as the annual acceleration in March over Kaskawulsh glacier. However, data-driven approaches may encounter limitations when data density is very low. In this case, *a priori* information, if available, may help to constrain the time-series. The regularization term can be modified to include a model, similar to López-Quiroz et al. (2009). An example is given in Eq. C1. By doing so, the inversion solves both the temporal closure and a parametric




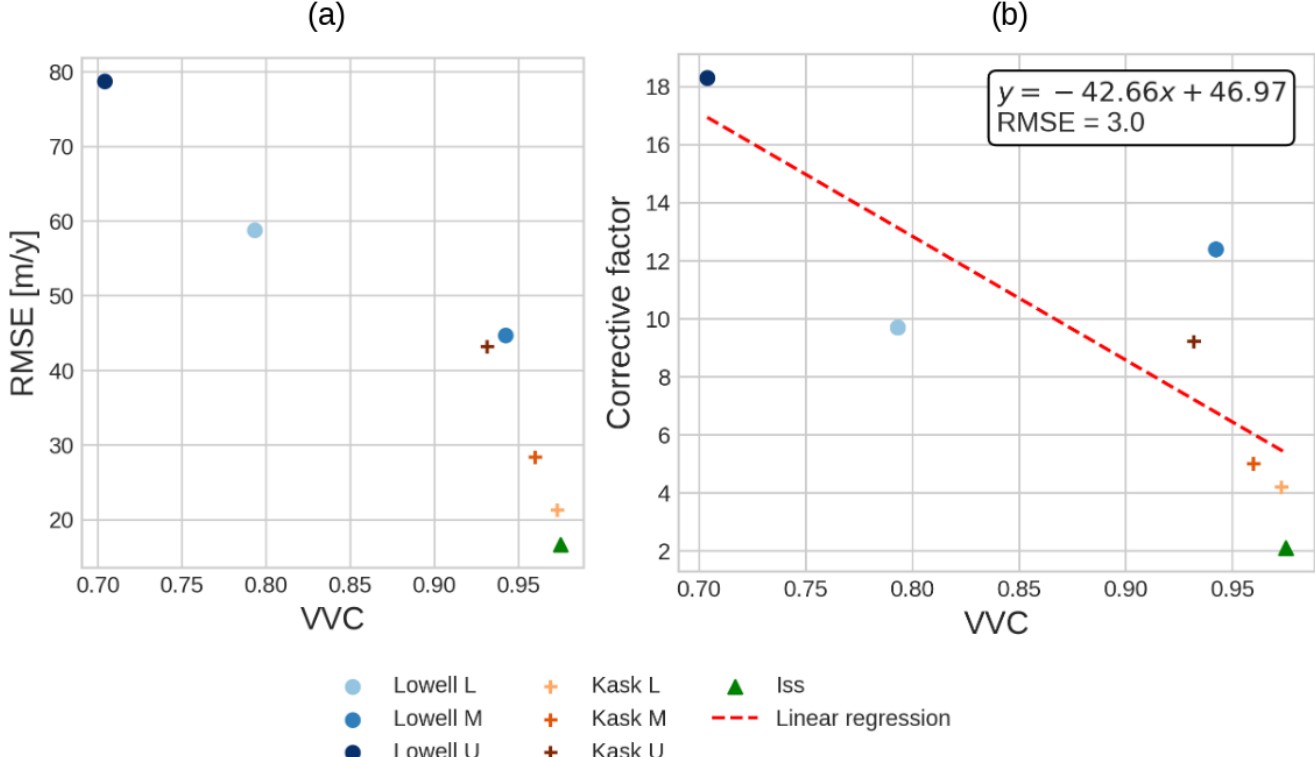

**Figure 12.** (a) Scatter plot of VVC and RMSE over the seven stations analyzed in this study, from 2013 to 2022. The RMSE corresponds to the comparison between TICOI results and GNSS data, while the VVC is an indicator of the temporal coherence of the directions of the TICOI results. (b) Correction factor that should be applied to the theoretical 95% confidence intervals so that they include both the TICOI estimate and the GNSS data.

regression problem. For more flexibility, it may also be possible to use a dictionary of functions (Riel et al., 2021; Hetland et al., 2012).

TICOI can be applied in a nearly automatic way, at a reasonable computational cost. We proposed a general strategy to automatically select the regularization coefficient. Moreover, the computational time of the TICOI processing chain including
loading, pre-processing, inversion and saving is about 0.1 seconds per pixel by using 32 CPUs (on a Intel(R) Xeon(R) Gold 6426Y with a processing rate of 3.3 GHz), for datasets containing 80,000 layers in time (corresponding to the period 2013-24). This means that the processing over a region of 100 km x 100 km requires around 19h. Note that the processing time per pixel scales with the size of the data in time, and that the 2013-24 period has high density of measurements compared to previous years. This computation time remains affordable at the regional scale, but may be more limiting at the global scale.
For application at global scale the computational time could be reduced, for instance, by taking advantage of GPUs, or by reducing the number of input data by using a stricter outlier filter.





With these two points in mind, it appears relevant to apply TICOI at a large scale. First, it reduces the size of the data in comparison with raw image-pair velocities by removing redundant information. On average, for our study sites, the size of the data is reduced by a factor 100. Second, it produces regularized velocity time-series, with relevant quality indicators. This would make the integration of sub-annual velocities in numerical models much more affordable.

## 6 Conclusion

To derive sub-annual velocity variations over glaciers, we propose an operational Python package, called TICOI, based on the temporal closure principle. This package fuses multi-temporal and multi-sensor image-pair velocities computed by different processing chains to generate regularized velocity time-series (i.e., sampled at regular time steps).

We improved previous methods based on two strategies: (1) a regularization term based on a coarse initial estimate, which enhances the resilience of the temporal closure inversion to abrupt non-linear changes; and (2) an iteratively reweighted approach, which automatically detects temporal decorrelation. TICOI is entirely data-driven, making it applicable to any glacier dynamics. The validation of TICOI results using GNSS data highlights an improvement in RMSE and KGE of around 50% in comparison with both the raw image-pair velocities and a rolling median. Furthermore, TICOI implies a change in paradigm by providing the ability to retrieve monthly velocities using annual image-pair velocities only. This could be especially useful for slow moving areas, where annual image-pair velocities may be of better quality than image-pair velocities with short temporal baselines. Moreover, TICOI can be used to combine datasets from different processing chains. This has the potential to reduce the uncertainty in the upper part of glaciers, such as in the accumulation area, where image-pair velocities are more noisy.

We recommend to use three criteria to assess the quality of the retrieved velocity series: (1) the VVC, a proxy of the temporal coherence of the direction; (2) the number of contributing image-pair velocities, and (3) a 95% confidence interval derived from the *a posteriori* covariance matrix. The application of TICOI provides velocity time-series with an unprecedented temporal resolution. On the Lowell glacier (Yukon, Canada), we are able to observe that summer velocities near the terminus started to increase five years before the surge. On Kaskawulsh glacier (Yukon, Canada), we are able to resolve velocity peaks in March in a very localized part of the lower ablation area.

Finally, the TICOI workflow offers reasonable computational time for application at the regional scale. The code is open-source and can be applied to any datasets and regions. This paves the way for the integration of a wide range of image-pair velocities and the production of standardized post-processed sub-annual velocity time-series.

*Code and data availability.* The TICOI package will be available on https://github.com/ticoi/ticoi once the paper has been accepted. ITS_LIVE data are available on https://its-live.jpl.nasa.gov/



*Author contributions.*   LCh, AD, FB, and RM designed the study. LCh, LG, AD, and FB proposed the methodological improvements. LCh, LG and NL worked on the python package with advice from AD. LCh generated the velocity time-series and analyzed the results with feedbacks from AD, FB and RM. LCo and CD provided processed GNSS data over Yukon, and helped with analysis of the results. LCh carried out the post-processing of Yukon GNSS data. NM provided processed GNSS data over western Greenland. LCh led the writing, and all the authors contributed to it.

*Competing interests.*   The author declare no competing interest.

*Acknowledgements.*   LCh acknowledges support from the Centre National d'Etudes Spatiales for her postdoctoral fellowship. AD and LCh acknowledge support from the French Programme National de Télédétection Spatiale (PNTS). Lei Guo thanks to the China Scholarship Council (No. 202306370154) for his scholarship. Luke Copland and Christine Dow thank the Natural Sciences and Engineering Research Council of Canada, Polar Continental Shelf Program, Canada Foundation for Innovation, Ontario Research Fund, and New Frontiers Research

Fund for funding to purchase and operate the Yukon GNSS stations, and the staff of Kluane Lake Research Station and graduate students from University of Ottawa and University of Waterloo for assistance and support in the field. We acknowledge Antoine Rabatel for his help in designing the study and his relevant feedbacks, Ghislain Picard and Fabien Gillet-Chaulet for their advice concerning the code optimization and the regularization term. We acknowledge the use of ChatGPT and DeepL for assistance in rephrasing and proofreading some sentences in the manuscript.



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



**Appendix A: Demonstration of the *posteriori* covariance matrix formula**

We demonstrate how the *posteriori* covariance matrix formula (eq. 11) has been derived. First, the analytic solution of a least square problem with a Tikhonov regularization is:

$$\hat{X} = N^{-1}A^{T}WY \tag{A1}$$

with $N = A^{T}WA + \lambda\Gamma^{T}\Gamma$

The *posteriori* covariance matrix is:

$$\Sigma_{\hat{X}} = \left[\frac{\partial\hat{X}}{\partial Y}\right]\Sigma_{\hat{Y}}\left[\frac{\partial\hat{X}}{\partial Y}\right]^{T} \tag{A2}$$

It corresponds to:

$$\Sigma_{\hat{X}} = N^{-1}A^{T}W\Sigma_{\hat{Y}}WAN^{-1} \tag{A3}$$

**Appendix B: Sensitivity analysis and automated choice of the hyperparameters**

**B1 Spatio-temporal filter**

We compared the performance of five types of spatio-temporal filters: Savitzky–Golay, Gaussian, Locally Weighted Scatterplot Smoothing (LOWESS), and median. The choice of filter results in a standard deviation in RMSE of about 2.6 m yr$^{-1}$ on average (i.e., 8% of the averaged RMSE), except for the GNSS station Lowell L, where the LOWESS filter produces an RMSE approximately 170 m yr$^{-1}$ higher than the other filters (Tab. B1).

LOWESS is a non-parametric moving regression that fits a model to the k nearest points (Derkacheva et al., 2020), which tends to over-smooth data during periods with low observation density. For example, the LOWESS solution flattens the surge peak of Lowell L, due to the limited number of observations available during that time (Fig. B2b).

We note that both the LOWESS and median filters can provide slightly better results for non-surge type glaciers, with improvements ranging from 0.8 to 4.5 m yr$^{-1}$ (i.e., 3% to 10%). However, they can also lead to over-smoothing (Fig. B2a) and require 1.5 times more computational time. Therefore, we recommend using the Savitzky–Golay filter, which offers a good balance between computational efficiency and accuracy in general scenarios.

**B2 Solver**

We compare four differents solvers: the Least Square solver (LS), LSMR, LSMR with an initialisation and LSQR. The RMSE between GNSS and TICOI time-series are really similar among GNSS stations. However, the computational time of the LS



|  | savgol | gaussian | median | LOWESS | std |
|---|---|---|---|---|---|
| **Lowell L** | **61,4** | 76,7 | 72,9 | 229,6 | 70,7 |
| **Lowell M** | 43,5 | 46,1 | 36,5 | **31,7** | 6,0 |
| **Lowell U** | **85,7** | 86,0 | 88,4 | 87,4 | 1,0 |
| **Kask L** | 22,3 | 23,8 | 20,1 | **20,0** | 1,8 |
| **Kask M** | 26,0 | 27 | **25,2** | 25,4 | 0,8 |
| **Kask U** | 42,2 | 46,1 | **37,7** | 39,7 | 3,6 |
| **Computional time [s]** | 67.8 | 67.2 | 69.4 | 101.6 | |

**Table B1.** Comparison of RMSE of the velocity magnitude between GNSS and TICOI time-series in m yr$^{-1}$ for different spatio-temporal filters (columns) and GNSS stations (rows). The averaged computional time betwwen all the GNSS stations are given in the last row in s. The filter svagol corresponds to the Savitzky–Golay filter. The LOWESS filter correspond to the statsmodel.nonparametric implementation and the Savitzky–Golay to scipy.signal implementation. The fraction used for the LOWESS filter is $60/N$ with $N$ the number of observations over the period. The temporal window of the Savitzky–Golay, median and gaussian filters are of 90 days. We note that the better performance obtained by the median and LOWESS filters on Lowell M is mainly due to the absence of GNSS data during the maximum of the surge.

is twice larger than for the other solvers. Therefore, we recommend using LSMR, LSMR_ini or LSQR for a question of
computational time.

|  | LSMR | LSMR_ini | LSQR | LS |
|---|---|---|---|---|
| **Lowell L** | 61.16 | 61.39 | 61.11 | 61.13 |
| **Lowell M** | 46.0 | 45.99 | 46.0 | 46.0 |
| **Lowell U** | 86.24 | 86.24 | 86.24 | 86.25 |
| **Kask L** | 20.9 | 20.89 | 20.9 | 20.91 |
| **Kask M** | 25.03 | 25.03 | 25.03 | 25.04 |
| **Kask U** | 40.12 | 40.15 | 40.13 | 40.14 |
| **Comptutional time [s]** | **40.8** | **48.8** | **43.0** | **85.2** |

**Table B2.** Comparison of RMSE of the velocity magnitude between GNSS and TICOI time-series in m yr$^{-1}$ for different solvers (columns) and GNSS stations (rows). The average computional time between all the GNSS stations are given in the last row in s. The solver Least Square (LS) corresponds to the function lstsq of numpy.linalg, the solvers LSMR and LSQR are respectively the functions lsmr and lsqr of scipy.linalg and scipy.sparse. LSMR_ini corresponds to the solver LSMR with an initialisation, corresponding to the spatio-temporal filtering observations $X_0$ defined in section 2.2.2.



## B3 Strategy to delete outliers

We compared two strategies for outlier removal: the Modified Z-score (Mzscore), which filters out values more than 3.5 NMAD away from the median of the entire time period, and the Median Angle (MA), which removes observations deviating by more than 45° from the median vector (Charrier et al., 2022a). We strongly advise against using the Mzscore for surge-type glaciers.
For non-surge glaciers, the improvement is at most 2 m yr$^{-1}$ compared to a strategy that does not filter outliers, largely because the Tukey biweight function, which is used, already acts as an outlier filter. Nevertheless, we recommend applying at least the MA filter to reduce the number of observations input into TICOI.

|  | **median_angle** | **mz_score** | **no_delete_outliers** |
|---|---|---|---|
| **Lowell L** | **58.79** | 711.44 | 59.15 |
| **Lowell M** | **44.71** | 511.52 | 44.8 |
| **Lowell U** | **78.72** | 96.74 | 81.72 |
| **Kask L** | **21.27** | 23.2 | 21.93 |
| **Kask M** | 28.37 | **25.46** | 27.36 |
| **Kask U** | 43.15 | **41.25** | 43.2 |
| **Iss** | **12.53** | 15.1 | 15.4 |

**Table B3.** Comparison of RMSE of the velocity magnitude between GNSS and TICOI time-series in m yr$^{-1}$ for different strategy to delete outliers (columns) and GNSS stations (rows). The minimal values are displayed in blod.

## Appendix C: Simulated data

### Synthetic instantaneous velocity and position time-series

The synthetic instantaneous velocity is taken as: $v(t) = a + b\sin(\frac{2\pi}{T}t) + c\cos(\frac{2\pi}{T}t)$ with $T = 365.25$ as in Greene et al. (2020). To make sure that the coefficients $a, b$ and $c$ represent well the data, instead of an arbitrary choice, these coefficients are estimated by an IRLS inversion by adding a regularization term corresponding to a displacement model with the coefficients $a, b$ and $c$ as parameters as performed in López-Quiroz et al. (2009). The corresponding system of equations is given in equation C1. The system is solved for Sentinel-2 data on the point represented in blue in Charrier et al. (2022c) Fig S1. The coefficients
are found to be: $a = -0.49$, $b = -0.0788$ and $c = 0.018$.

$$\begin{bmatrix} 1 & 1 & 0 & \dots & 0 & 0 & 0 & 0 & 0 \\ \vdots & \vdots & \vdots & \vdots & \vdots & \vdots & \vdots & \vdots & \vdots \\ 0 & 0 & 0 & \dots & 1 & 0 & 0 & 0 & 0 \\ \lambda & 0 & 0 & \dots & 0 & -[\tau_{1+\Delta\tau} - \tau_1] & -\frac{T}{2\pi}[\sin(\frac{2\pi}{T}\tau_{1+\Delta\tau}) - \sin(\frac{2\pi}{T}\tau_1)] & -\frac{T}{2\pi}[\cos(\frac{2\pi}{T}\tau_{1+\Delta\tau}) - \cos(\frac{2\pi}{T}\tau_1)] & 1 \\ \vdots & \vdots & \vdots & \vdots & \vdots & \vdots & \vdots & \vdots & \vdots \\ 0 & 0 & 0 & \dots & \lambda & -[\tau_{1+n\Delta\tau} - \tau_{1+(n-1)\Delta\tau}] & -\frac{T}{2\pi}[\sin(\frac{2\pi}{T}\tau_{1+n\Delta\tau}) - \sin(\frac{2\pi}{T}\tau_{1+(n-1)\Delta\tau})] & -\frac{T}{2\pi}[\cos(\frac{2\pi}{T}\tau_{1+n\Delta\tau}) - \cos(\frac{2\pi}{T}\tau_{1+(n-1)\Delta\tau})] & 1 \end{bmatrix} \begin{bmatrix} \hat{d}_{\tau_1,\tau_{1+\Delta\tau}} \\ \vdots \\ \hat{d}_{\tau_{1+(n-1)\Delta\tau},\tau_{1+n\Delta\tau}} \\ a \\ b \\ c \\ d \end{bmatrix} = \begin{bmatrix} d_{\tau_1,\tau_{1+2\Delta\tau}} \\ \vdots \\ d_{\tau_{1+(n-1)\Delta\tau},\tau_{1+n\Delta\tau}} \\ 0 \\ \vdots \\ 0 \end{bmatrix}$$



(C1)

where $T$ is the period of the sinusoidal signal. $a, b, c, d$ are the coefficients of the model.

The position time-series is defined as the integral of the instantaneous velocity.

**Selection of acquisition dates**

Then, we randomly select the acquisition dates in a list of dates ranging from the 1 January 2015 and 31 December 2020, every 5 days. By doing so, some dates between the 1 January 2015 and 31 December 2020 will not be included in the simulated dataset. It represents the effect of clouds: the pixels covered by clouds will be systematically rejected.

**Noise**

For each acquisition date, we add a Gaussian noise to the position value. This accounts for the fact that the noise depends mainly on the image texture (clouds, snow, crevasses, etc.). Therefore, the noise of each displacement is the sum of the noises of each date of the pair.

**Image-pair velocity**: We randomly select a temporal baseline between 5 and 400. Then, we compute image-pair velocity by taking the difference between the position at the second date of acquisition and the first date of acquisition.





**Figure A1.** Distribution of the errors estimated by comparing measured displacements from remote sensing images and GNSS displacements. The left column shows errors in East/West ($D_x$) and North/South ($D_y$) displacements according to the temporal baseline. The right column shows the distribution of this error. Skewness is a measure of the symetry of a distribution, a value of 0 indicates a symetric distribution. Kurtosis refers to the degree of "tailedness" of a distribution relative to a normal distribution. Strong kurtosis (>3) reveals heavy-tailed distribution.

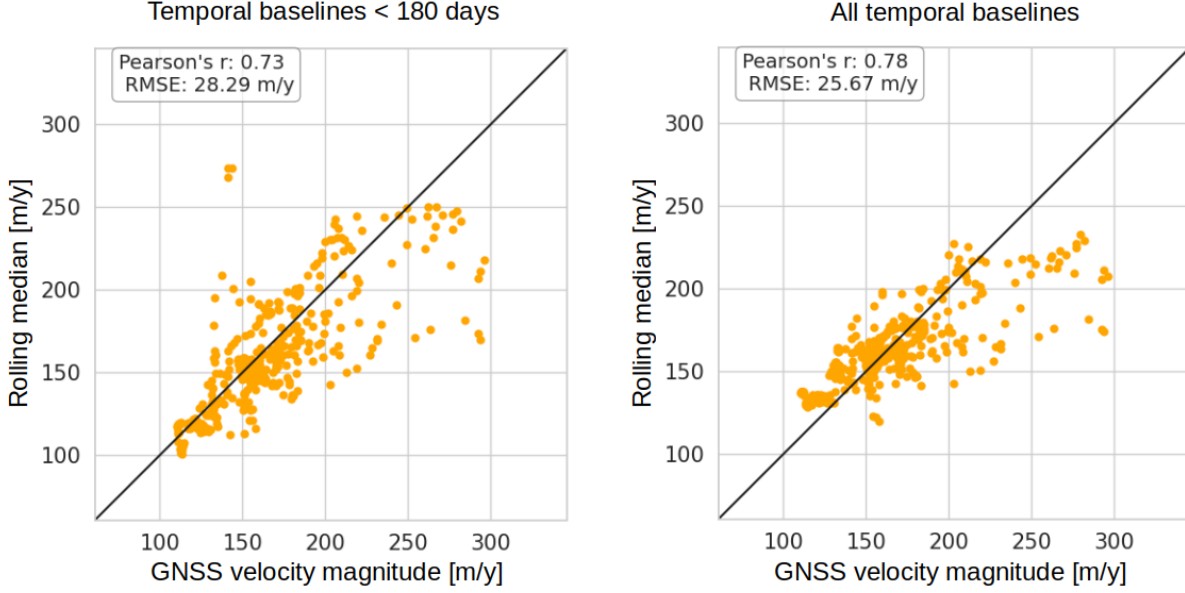

**Figure A2.** Scatterplot of GNSS velocity magnitude and 30 days rolling median applied to velocity magnitude observations, on the left with a temporal baseline lower than 180 days and on the right with every temporal baseline. The RMSE is better while using all temporal baselines, but there is a clear underestimation for velocities larger than 180 m yr$^{-1}$.



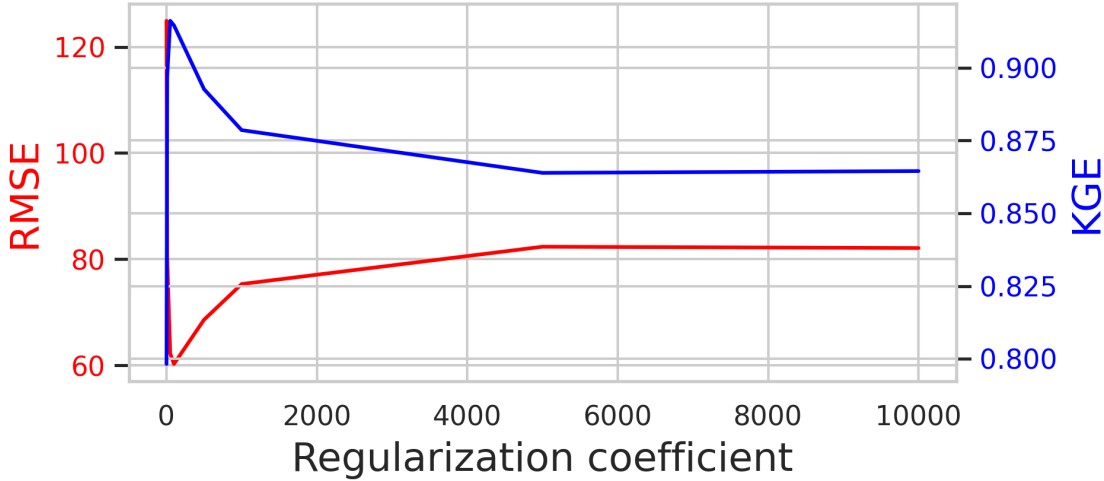

**Figure B1.** Evolution of the RMSE and KGE of TICOI results with respect to GNSS data for station Lowell L, where a surge occurs. When the coefficient increases the acceleration of the TICOI estimations tend to be close the initial guess of acceleration, which in this case slightly oversmooths the peak of the surge. This is why the RMSE and KGE reach a plateau after a coefficient of around 5000.

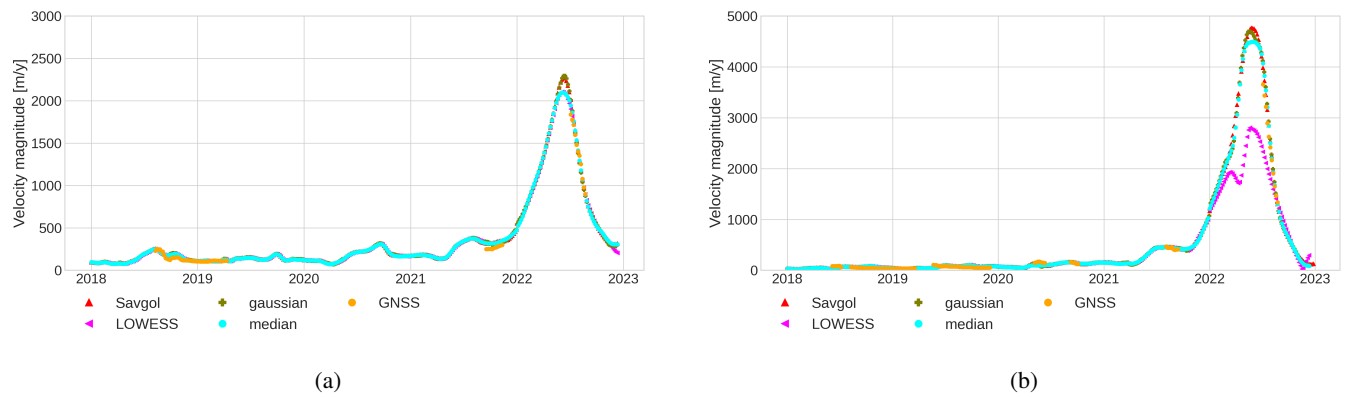

**Figure B2.** (a) Visual comparison of the filters for the point Lowell M. (b) Visual comparison of the filters for the point Lowell L.