# Peer review of "TICOI: an operational Python package to generate regular glacier velocity time series"

_EGUsphere, 2024_

## Referee Comment (RC2)

**Review of - *TICOI: an operational Python package to generate regularized glacier velocity time series***

By *Laurane Charrier, Amaury Dehecq, Lei Guo, Fanny Brun, Romain Millan, Nathan Lioret, Luke Copland, Nathan Maier, Christine Dow, and Paul Halas*

The authors describe a new open source python package, TICOI, for the creation of regularly temporally spaced velocity measurements from large velocity pair stacks. The package builds on some of the authors recent work with 'overdetermined system' inversions, and adds an interpolation step to create regularly spaced outputs in time. The manuscript is well written overall, and the package described will be useful to the community. I have included a number of comments below, but these are all relatively minor and should not require any major reorganisation of the manuscript. Overall, I recommend publication in TC following minor revisions.

On the whole, this will be a substantial improvement to our treatment of ice velocity timeseries, and to their usability by wider audiences (the uneven temporal spacing is always a source of confusion). There are still a number of open questions about the best way to treat uncertainties in this data but I cannot fault the authors for not resolving these here. This package also makes for an excellent baseline for future development to build off, particularly as the authors have made sure it is open source and adequately documented.

I attach more detailed comments here:

L1 I understand the meaning but this is slightly awkward wording, maybe something more precise than "glacier mass redistribution and future geometry"?

L2 in open-source -> open source

L5 This point on numerical models could be elaborated on, for instance there are implications for SLR predictions through the ways ice velocity is used in ice sheet models – not always accounted for in final uncertainties.

L9 perhaps 'evaluated against' rather than 'validated using'?

L12-13 'also proves to be able to' -> 'can also'/'In addition, TICOI can'

L13 'under certain conditions' perhaps? I imagine it cannot do this reliably in all cases.

L14 'regularization' -> 'harmonization' / 'improving the intercompatibility'?

L14 It might be interesting to state what period you think might be optimal in most cases – daily, 5 days, weekly, monthly?

L16 + In the intro discussing the importance of ice velocity a quick mention of the hazard implications would also be good – it can be key to understanding dangerous surges (particularly where lake dams are formed; e.g. Singh et al., 2023 https://doi.org/10.1016/j.scitotenv.2023.161717; Beaud et al., 2022 https://doi.org/10.1016/j.geomorph.2019.106957), forecasting glacier detachments (e.g. Kaab et al. 2021 https://doi.org/10.5194/tc-15-1751-2021; Gilbert et al., 2018

https://doi.org/10.5194/tc-12-2883-2018), and understanding volcanic influences (e.g. Martin et al., 2025 https://doi.org/10.1017/jog.2024.107) among others. It is in this domain that challenges with data quality and uncertainty quantification can be particularly problematic so this can help you make the case for the value of TICOI. It is a step towards integrating satellite data in early warning systems for some of these processes – though this is perhaps still some way off.

L57 'associated quality indicator' is a little vague here – is this because it is a 'relative quality score' rather than 'absolute uncertainty'? A few more words to clarify would help.

L120 Specifying what 'a priori knowledge of the data quality' means here might be useful?

L140 Not sure 'relevant' is what you mean (?)

L143-144 The first point here is not clear, could you rephrase? It is not clear why this would prevent comparison as written. We don't typically have instantaneous velocities.

L144-145 for 2), this could be an argument against interpolation unless uncertainties are properly captured, as a regular interval interpolated from long/short baseline images would have different expected error profiles

L145 For 3) presumably this could be solved via interpolation also?

L148-149 This assumption will in fact be wrong over any time window, not just long ones (glacier velocity is never truly constant over any meaningful timescale, though the approximation may be closer in some cases). And seasonal variations/surges are not the only processes involved.

L155 I understand this might be covered in other paper, but does this account for low/no decorrelation over stable ground compared to the ice?

L159-160 Can you be more specific here rather than referring to 'the above'?

L162-169 As far as I understand this is assuming that decorrelation means bad data? Can you state this here?

L177 'affordable' -> 'valid'

L179 Do we not have some a priori information in all cases from our understanding of ice physics? (granted, with a wide range) Could the details of a constraint be calibrated further from easily available glacier data (e.g. geometry)?

L189-196 I gather this requires looping through all pixel timeseries individually. Is there no way to vectorise the least squares inversion step so that this can be run on the 3D cube in one operation? Would this then be gated by memory usage? This vectorisation should be fairly straightforward to do for the interpolation step if not already implemented.

L198-207 The VVC will penalise areas with real temporal variation in flow direction, right? E.g. parts of ice front or glacier convergence areas.

L210 Could you discuss a little more the choices of uncertainty metric here? In particular some of the potential weaknesses of the choice and reasons for excluding other common values (such as stable ground displacement and x-correlation signal to noise ratio).

In reality we have two types of uncertainty in feature tracking:

1) What is the likelihood that this displacement value is actually representing the real motion process (which you raise above with the discussion about decorrelation). If it is not, then there is no information in the resulting value and an ideal processing workflow would exclude it. This is similar to 'accuracy' but more of a binary real/not real.

2) For a displacement value representing the real motion process (i.e. 'real' in the above), how precise is this? This will be affected by warping of features, partial decorrelation, the subpixel algorithm choice, image georeferencing error, etc.

It seems that the approach here perhaps blends these two together – this can be fine as they are hard to separate out in many cases. It would be interesting to have a little more discussion about this and the choices made.

L294-294 Can you say a little more about this 'increased iteratively according to the normalized displacement coherence' – i.e. the window size increases if the noise (calculate from NDC) is above a given threshold. It is an important factor of AutoRIFT to understand, as the uneven window sizes leads to uneven smoothing (and thus error/noise) through space for a single image pair.

L306-308 Why was this necessary rather than reprojecting the NS and EW components?

L349 Can you show that this doesn't sometimes reject real long baseline data in areas where the shorter baseline data is particularly noisy?

L362 I like the KGE, but it is rarely used in this field so far. It might be useful to say in a little more detail here why you think it is a useful indicator and cite a relevant paper for more info (e.g. Gupta et al. 2009 https://doi.org/10.1016/j.jhydrol.2009.08.003). Give some idea of what a 'good' KGE might be (it is not a 0-1 quality metric as some might assume from a quick read of this as is).

Personally I might also look at the three KGE components separately also as the response can be dominated by one of them.

L369-375 The numbers are confusing in here. You use percentages in places I am not clear why – RMSE and KGE are not usually given in percent. Does "a reduction in RMSE from 9 to 69%" mean a reduction 'from 9 to 0.69', or this a typo and you mean 'from 69 to 9'? Please clear this up.

Tab 1: Interestingly, this seems to show that moving median can be decent for some glacier types. I wonder if the median split by temporal baseline might do even better in some cases (though of course not picking up abrupt changes as TOCOI can, but will always be much computationally faster).

L405 – This paragraph should capture the fact that flow direction will not be constant across all of a glacier, and some areas may have real processes leading to low VVC even without noise. The confluence of two glaciers is one such place (i.e. varying ice flux from one or the

other will shift exact position of confluence and the local flow direction). This looks particularly of concern here for the terminus (of special interest, but also with naturally low VVC in many cases).

Figure 7 – I am a little confused by the visualisation here – is this showing that a large majority of estimate values are within 9cm of the true value? Seems too high, so I am perhaps misunderstanding.

Figure 8 – How do you know that the grey bars are 'correct'?

L465-466 Did you try this on the Lowell and Kaskawulsh case studies also? How well did you pick up the marked sub-annual variation in that case? I think section 4.6 likely oversells the generalisability of this to all cases.

L484-485 this should be 1m and 3m respectively for 0.1pix

L527 As currently set up this is fully parallelised right? So for ~300 CPU could be run in ~2hr? Seems feasible to run at least all non-ice sheet data with a large machine.

L530 Are there ways to exploit the spatial autocorrelation to reduce computational cost also?

L543 'entirely data-driven' – you mean it requires no a priori info/ model here right?

L544 KGE improvement can't really be capture in a percentage. From table 1 Kask L had a negative 260% change due to sign flip! Give absolute value increases for it.

L545 I agree this is a useful advance in the way this is done but I probably wouldn't describe it as a paradigm shift. The iterative procedure implemented in GIV can also do this (VWDV and Wickert 2022, https://doi.org/10.5194/tc-15-2115-2021), though almost certainly not as well – the iteration can be rather sensitive to outliers.

On a different note it would be nice to apply slightly more caution in the wording of this. Many areas have limited short baseline data due to high cloud cover (e.g. many HMA glaciers) but also large temporal variability within this period – it probably won't be very effective at reconstructing these. Including an assessment of the Lowell/Kask glaciers in the example above on this might help illustrate.

L555 instead of 'reasonable computational time' using an exact number here would be better.

Thanks again to the authors for this contribution and I look forward to testing TICOI myself.

-Max VWDV

---

## Author Response (AR1)

We thank Maximillian Van Wyk de Vries for his careful reading of our manuscript, his detailed comments and help in improving our manuscript. We have made all the changes suggested. Please find below our answers (in black).

The authors describe a new open source python package, TICOI, for the creation of regularly temporally spaced velocity measurements from large velocity pair stacks. The package builds on some of the authors recent work with 'overdetermined system' inversions, and adds an interpolation step to create regularly spaced outputs in time. The manuscript is well written overall, and the package described will be useful to the community. I have included a number of comments below, but these are all relatively minor and should not require any major reorganisation of the manuscript. Overall, I recommend publication in TC following minor revisions.

On the whole, this will be a substantial improvement to our treatment of ice velocity timeseries, and to their usability by wider audiences (the uneven temporal spacing is always a source of confusion). There are still a number of open questions about the best way to treat uncertainties in this data but I cannot fault the authors for not resolving these here. This package also makes for an excellent baseline for future development to build off, particularly as the authors have made sure it is open source and adequately documented.

L1 I understand the meaning but this is slightly awkward wording, maybe something more precise than "glacier mass redistribution and future geometry"?
We thanks the reviewer, and have modified the text as:
"Glacier flow velocity is a crucial observation as it controls the mass redistribution and future evolution of the geometry of a glacier."

L2 in open-source -> open source
changed everywhere

L5 This point on numerical models could be elaborated on, for instance there are implications for SLR predictions through the ways ice velocity is used in ice sheet models – not always accounted for in final uncertainties.
We reformulated as: "This hinders our ability to understand flow processes such as basal sliding and surges, as well as the integration of these observations into numerical models. The latest could help to better constrain future projections of sea level rise." and also complemented the introduction as:

"Additionally, several recent methodological and modelling developments stress the need for precise and temporally resolved velocity products to infer basal conditions (Jay-Allemand et al., 2011; Goldberg et al., 2015) or near-future projection (Choi et al., 2023), for example using transient inverse methods (Goldberg et al., 2015; Choi et al., 2023). **This could help to better constrain future sea-level rise."**

L9 perhaps 'evaluated against' rather than 'validated using'?
Modified

L13 'under certain conditions' perhaps? I imagine it cannot do this reliably in all cases.
Modified to : In addition, TICOI can retrieve monthly velocity using annual image-pair velocities only, when there is enough temporal redundancy.

L14 'regularization' -> 'harmonization' / 'improving the intercompatibility'?
Modified to harmonization.

L14 It might be interesting to state what period you think might be optimal in most cases – daily, 5 days, weekly, monthly?
We thanks the reviewer for this comment. We added in the abstract: In this article, we provide extensive examples of TICOI application on the ITS\_LIVE dataset and in-house velocity products, to generate monthly velocity time-series.
This is discussed further in the text.
Section 2.1.3 « By analyzing the Root Mean Square Error (RMSE) over stable areas, we have shown that the RMSE according to the temporal sampling has an asymptotic behavior which converges after around 30 days for glaciers with medium average velocity ($\sim$100 to 200 myr−1 ) (Charrier et al., 2022a, b). »

L16 + In the intro discussing the importance of ice velocity a quick mention of the hazard implications would also be good – it can be key to understanding dangerous surges
We added the references in the introduction.

L57 'associated quality indicator' is a little vague here – is this because it is a 'relative quality score' rather than 'absolute uncertainty'? A few more words to clarify would help.
There is both absolute uncertainty and a relative quality score.
modified to with an associated uncertainty and relative quality indicator.

L120 Specifying what 'a priori knowledge of the data quality' means here might be useful?
We added: "(e.g., image correlation score or velocity in stable areas \citep{gardner2018increased})"

L140 Not sure 'relevant' is what you mean (?)
Changed to « However, we need regular time series to study glacier dynamics \citep{charrierfusionisprs}. »

L143-144 The first point here is not clear, could you rephrase? It is not clear why this would prevent comparison as written. We don't typically have instantaneous velocities.
We added more explanation : « velocities with different temporal sampling are not comparable because they correspond to the average of the instantaneous velocity over different time intervals (e.g., annual image pair-velocities are closed to the annual average of the instantaneous velocity whereas short temporal baseline velocity are close to the instantaneous velocity)

L144-145 for 2), this could be an argument against interpolation unless uncertainties are properly captured, as a regular interval interpolated from long/short baseline images would have different expected error profiles
Yes, this problem is alleviated by interpolating the cumulative displacement time-series. The error in displacement does not depend on the temporal baseline.

L145 For 3) presumably this could be solved via interpolation also?
Yes definitely.

L148-149 This assumption will in fact be wrong over any time window, not just long ones (glacier velocity is never truly constant over any meaningful timescale, though the approximation may be closer in some cases). And seasonal variations/surges are not the only processes involved.
We thank the reviewer for this recommendation, and modify the text to : « This assumption is most of the time inaccurate, particularly over long temporal windows or in cases where the glacier exhibits surge behaviour or seasonal variations \citep{charrier2022grsl}. »

L155 I understand this might be covered in other paper, but does this account for low/no decorrelation over stable ground compared to the ice?
We agree with the reviewer : stable area analysis may not be as much impacted by temporal decorrelation. However, the temporal sampling mainly impact random errors. When the temporal sampling dt increase, the instantaneous signal is averaged over a larger time span. Random errors in displacement in m are divided by dt when converted to m/y.

L159-160 Can you be more specific here rather than referring to 'the above'?
We modified the text to « An IRLS with a first order Tikhonov regularization term performs poorly in some extreme cases, such as temporal decorrelation or abrupt non-linear changes, especially when there is few image-pair velocities. ».

L162-169 As far as I understand this is assuming that decorrelation means bad data? Can you state this here?
Modified to : « " Robust LS regression, like IRLS using Tukey's bi-weight function, helps to reduce the effect of outliers in case of random errors (Liang et al., 2020; Charrier et al., 2022b) but may be inefficient for systematic errors. For example, when temporal decorrelation occurs, the measured displacement is systematically close to 0, instead of the true glacier velocity, which results in a heavy-tailed distribution of errors with a strong kurtosis." »

L177 'affordable' -> 'valid'
Modified.

L179 Do we not have some a priori information in all cases from our understanding of ice physics? (granted, with a wide range) Could the details of a constraint be calibrated further from easily available glacier data (e.g. geometry)?
We thanks the reviewer for this question. It is pretty easy to add a constrain based a model inside TICOI, as discussed in the discussion (section Large scale application). However, we think that the community should be really careful while doing that for two reasons :1) this will add a sensitivity to physical assumptions, often necessary to model glacier dynamics. For example, it could be possible to constrain the along-flow strain rates, as developed in GLAFT (Zheng et al., 2023). However, this requires physical assumptions, for example Zheng et al wrote « we assume no basal slip in this calculation, which may not be physically realistic for Kaskawulsh Glacier and likely yields an overestimated recommendation. » This may not be problematic for a quality metric, but it can introduce biases if this is used to constrain velocity time series. 2) it will add a sensitivity to other datasets quality. For example, the constrain can be based on the mass conversation, however it requires ice thickness data, rate of elevation change, surface mass balance and density, which are not necessarily available with a high quality worldwide.

L189-196 I gather this requires looping through all pixel time series individually. Is there no way to vectorise the least squares inversion step so that this can be run on the 3D cube in one operation? Would this then be gated by memory usage? This vectorisation should be fairly straightforward to do for the interpolation step if not already implemented.
We also wondered that, and tried to vectorize the process for the whole cube. However, we faced some problems. The time dimension of X (estimated displacements) is different from one pixel to another, because the filtering of outliers is different. However, it should be easier to vectorize the interpolation. We never tried it because this is not the most time consuming time step. We add this idea to the list of possible improvement for ticoi. Thanks.

L198-207 The VVC will penalise areas with real temporal variation in flow direction, right? E.g. parts of ice front or glacier convergence areas.

The VVC is only used to select an optimal regularisation coefficient by looking at the inflection point of the VVC curve. If there is a real change in the glacier flow direction (e.g. around a rift, over parts of the ice front or glacier convergence areas), the solution will still converge to a plateau, which has a lower value than for the case of a constant flow direction. In any case, it is advisable to calculate the VVC over relatively large areas (such as a whole massif) in order to be more robust. It's important to emphasise that TICOI never restricts the direction of flow.

We added in the text, at the end of section 2.4:« If there is a real change in velocity direction over time, the VVC curve will converge before 1, but the curve will still have an inflection point. For better robustness, it is advisable to calculate the VVC over a relatively large area. »

L210 Could you discuss a little more the choices of uncertainty metric here? In particular some of the potential weaknesses of the choice and reasons for excluding other common values (such as stable ground displacement and x-correlation signal to noise ratio).
In reality we have two types of uncertainty in feature tracking:
1) What is the likelihood that this displacement value is actually representing the real motion process (which you raise above with the discussion about decorrelation). If it is not, then there is no information in the resulting value and an ideal processing workflow would exclude it. This is similar to 'accuracy' but more of a binary real/not real.
2) For a displacement value representing the real motion process (i.e. 'real' in the above), how precise is this? This will be affected by warping of features, partial decorrelation, the subpixel algorithm choice, image georeferencing error, etc.
It seems that the approach here perhaps blends these two together – this can be fine as they are hard to separate out in many cases. It would be interesting to have a little more discussion about this and the choices made.

We thank the reviewer for highlighting this point. We agree that the assessment of errors is a key issue. The first type of error mentioned by the reviewer is also known as systematic error, while the second type corresponds to random errors. Both can be distinguished by looking at precision versus accuracy. Systematic errors are characterised by low accuracy, while random errors are more likely to have low precision (see figure below).  However, if there are random and systematic errors within the same dataset, it may be difficult to separate them.  The output uncertainty should ideally take both into account.

For glacier velocity, there is no ideal uncertainty metric. The latter is often based on stable ground displacement. The central tendency (mean, median) can be considered as the accuracy of the velocity map, while the variation around the central tendency (standard deviation, median absolute deviation) is often associated with the precision (Zheng et al., 2023 ; Dehecq et al., 2015, Paul et al., 2017). Note, that the central tendency is often removed in each of the velocity maps, to compensate remaining errors in image georeferencing. Then, one can consider robust metrics (median and Median Absolute Deviation) to mitigate the effects of large and isolated errors (which often correspond to systematic errors). In the literature, uncertainty is often based on the MAD of stable regions (Dehecq et al., 2015), which is probably close to random errors. Recently, Zheng et al. (2023) suggested using multivariate kernel density estimation of stable ground displacement to more easily separate correct and incorrect matches (in other words, random and systematic errors). This common way of using stable ground displacement to assess uncertainty is included in TICOI because we propagate errors based on stable ground displacement through the LS inversion. However, we agree that we could add a module to calculate errors based on stable ground directly on the TICOI results. We added the MAD calculation in TICOI, and in manuscript. We contacted Whijay Zheng to adapt GLAFT to NetCDF files (for now, the function is applied on Geotiff files, but it should not be super difficult to adapt it).

However, these metrics underestimate errors, because stable areas may not represent the glacier texture (Zheng et al., 2023, Altena et al., 2022). Therefore, quality indicators are often provided in the literature. This can be the cross-correlation signal to noise ratio as mentioned by the reviewer, along-flow shear strain rate developed in GLAFT by Zheng et al., 2023, or a metric based on the temporal coherence of the direction (Dehecq et al., 2015). Values of cross-correlation score or signal to noise ratio are available only if provided with the input dataset and cannot be calculated a posterio in TICOI. Therefore, this error metric is dataset dependent. In contrast, the Velocity Vector Coherence (VVC) can be calculated for any input dataset. But we do agree that it could interesting to add the possibility to use the along-flow shear strain rate developed in GLAFT.

Lastly, we add a final criterion: the number of image-pair velocities that have contributed to each estimation, which is also a common metric used in SBAS-like processing chains such as NSBAS : https://formater.pages.in2p3.fr/flatsim/products/Net.html.

We added more explanation in the discussion part of the manuscript:
-confidence intervals:
"Second, the underestimation of our confidence intervals could be caused by biases in the image-pair velocities, for example due to shadows or seasonal illumination changes \citep{lacroix_self-entrainment_2019}. The errors in the ITS_LIVE dataset are based on the standard error in component velocities relative to stable surface velocity; they characterize random errors. Therefore, our confidence intervals only account for random errors and not systematic biases."
-VVC and number of image-pair velocities :
With the current state of knowledge in velocity errors, we recommend relying on the VVC and number of contributed image-pair velocities. The VVC is a quality metric which characterize random errors, by analysing the temporal coherence of the direction. The number of contributed image-pair velocities indicated the robustness of the TICOI estimation: fewer than 100 image-pair velocities did not appear to provide sufficient constraint.
-NMAD over stable areas:
"Note, that the TICOI package also offers the possibility to compute the Normalized Median Absolute Deviation (MAD) over stable areas. This is a widely used and robust metric for characterizing random errors in glacier velocity fields. However, as previously demonstrated for ITS\_LIVE scene-pair velocities, such errors are often underestimated due to the differences in texture between glacier surfaces and stable ground. Moreover, the NMAD over stable ground do not capture the spatial variability in errors because they provide only one value for the entire scene at a given time. For example, the RMSE between TICOI and the GNSS is about 43 m yr$^{-1}$ on the upper of Kaskawulsh glacier, which is 20 m yr$^{-1}$ below the maximal NMAD obtained in the area (Fig. \ref{fig:stable_areas} b))."

We also added a few lines to describe other possible options in the discussion:
"To complement stable ground analysis, an alternative quality criterion has been proposed by \citet{zheng2023glacier}: the along-flow shear strain rate, which provides insight into the smoothness of the velocity solution. To enhance the flexibility of the package, this metric will also be included as an optional quality check."

[Figure]

L294-294 Can you say a little more about this 'increased iteratively according
to the normalized displacement coherence' – i.e. the window size increases if the noise
(calculate from NDC) is above a given threshold. It is an important factor of AutoRIFT to
understand, as the uneven window sizes leads to uneven smoothing (and thus error/noise)
through space for a single image pair.

We add more information on ITS_LIVE processing based on the reviewer comment : « The size of
the correlation window is increased iteratively according to a threshold on the Normalized
Displacement coherence, an indicator of the quality of the correlation \citep{gardner2018increased}
(i.e., there are uneven correlation window sizes which could lead to uneven smoothing and
uncertainties in space). »

L306-308 Why was this necessary rather than reprojecting the NS and EW components?

We added more explanation:

"This process requires reprojecting both the grid coordinates and the values of the EW and NS
velocity components. These velocity components are defined relative to the orientation of the grid,
representing the projection of the velocity vector along the grid's two axes. Consequently, it is
necessary to reproject the grid and recalculate the velocity vector projections along the new axes.
To achieve this, we first compute the coordinates of the start and end points of each velocity vector
in the new coordinate system.  We then calculate the difference between these coordinates along the
axes of the reprojected grid to obtain the new component values."

L349 Can you show that this doesn't sometimes reject real long baseline data in areas where
the shorter baseline data is particularly noisy?

If the automatic detection of temporal decorrelation was rejecting real long temporal baselines
where the shorter temporal baselines are noisy, this would lead to larger random errors.

As suggested by the reviewer before, we used the NMAD in stable areas of the results obtained
using TICOI and the automatic detection of temporal decorrelation. We added these lines in the
manuscript (section Robustness to temporal decorrelation)

"The NMAD over stable areas, a proxy of the precision of the method, slightly decreases with the use of the automatic detection of temporal decorrelation: the median NMAD is about 12.96 m yr$^{-1}$ without the automatic detection, against 12.19 m yr$^{-1}$ (Fig. \ref{fig:stable_areas} (a) and (b)). The median value over stable areas, a proxy of the accuracy, is about 19.02 and 18.13 m yr$^{-1}$ without and with the automatic detection respectively (Fig. \ref{fig:stable_areas} (c) and (d)). This highlights in increase in both accuracy and precision."

From this analysis, it seems that real long baseline data are rarely rejected.

[Figure]

L362 I like the KGE, but it is rarely used in this field so far. It might be useful to say in a little more detail here why you think it is a useful indicator and cite a relevant paper for more info (e.g. Gupta et al. 2009 https://doi.org/10.1016/j.jhydrol.2009.08.003). Give some idea of what a 'good' KGE might be (it is not a 0-1 quality metric as some might assume from a quick read of this as is).
Personally I might also look at the three KGE components separately also as the response can be dominated by one of them.

We thanks the reviewer for this recommendation. We added the reference, and this sentence : « The KGE values range between - infinity and 1. A perfect agreement between two time-series would lead to a KGE of 1, while poor agreement can lead to negative (up to infinite) values.»

L369-375 The numbers are confusing in here. You use percentages in places I am not clear why – RMSE and KGE are not usually given in percent. Does "a reduction in RMSE from 9 to 69%" mean a reduction 'from 9 to 0.69', or this a typo and you mean 'from 69 to 9'? Please clear this up.
We thanks with reviewer for highlighting this point. We removed percentage for the KGE, and added also absolute value for the RMSE, in addition of the percentage. A reduction in RMSE from 9 to 69% means that the RMSE is reduced by 9% in the least favorable case and 69% in the most favorable case.

Tab 1: Interestingly, this seems to show that moving median can be decent for some glacier types. I wonder if the median split by temporal baseline might do even better in some cases (though of course not picking up abrupt changes as TOCOI can, but will always be much computationally faster).
We agree with the reviewer that it will be interesting to carry on a inter-comparison exercise of post-processing approaches, in the future. The algorithms proposed so far (GIV, TICOI, LOWESS, etc.) have different trade-off between smoothing, and computationtime, different robustness against random and systematic errors. Probably some algorithms are more appropriate for some cases. It would be interesting to have a guideline on this.

L405 – This paragraph should capture the fact that flow direction will not be constant across all of a glacier, and some areas may have real processes leading to low VVC even without noise. The confluence of two glaciers is one such place (i.e. varying ice flux from one or the other will shift exact position of confluence and the local flow direction). This looks particularly of concern here for the terminus (of special interest, but also with naturally low VVC in many cases).
We agree with the reviewer, and added this line :
« Note that real changes in velocity direction can be expected over time in aras of variable flow, such as near confluences, glacier edges or terminus, and cause a low VVC that is not related to measurement error.»

Figure 7 – I am a little confused by the visualisation here – is this showing that a large majority of estimate values are within 9cm of the true value? Seems too high, so I am perhaps misunderstanding.
The values of the figure 7 only represent the percentage of correct confidence intervals, in the sense that they include both the estimated and the true value.

Modified to : « In controlled conditions, the 95\% confidence interval includes both estimated and true velocity (i.e. are correct) for more than 95\% of the estimation, except for low percentage of data and low noise where the confidence tends to be slightly underestimated (Fig.~\ref{fig:simulation}). » and « However, only 48\% of the confidence intervals include both the estimated and the GNSS velocities (i.e. are correct), which is much below the expected 95\%. On average, over the six GNSS stations, the percentage is 27\%."

The legend of the figure has been modified to: "Percentage of the estimated 95\% confidence intervals that include both the estimated and the true displacement values (i.e. the valid confidence intervals), using simulated data described in \cref{supp:simulated_data}."

Figure 8 – How do you know that the grey bars are 'correct'?
Line 1 :1 represent the « expected » values, GNSS data are equal to TICOI on this line. If the bars, representing the confidence intervals, intersect this line, it mean they include both the TICOI estimation and the expected value. If so, they are considered as correct.  This is the case for the grey bars.
We add more explanation in the caption :
Vertical grey bars correspond to the confidence intervals, which should intersect the red line 1:1 (i.e., encompass the true velocity value) if they are not underestimated. Underestimated confidence intervals are displayed in red, correct one are represented in grey.

L465-466 Did you try this on the Lowell and Kaskawulsh case studies also? How well did you pick up the marked sub-annual variation in that case? I think section 4.6 likely oversells the generalisability of this to all cases.

We tried in the lower part of the Kaskawulsh glacier, see Fig below. TICOI is also able to retrieve accurately monthly velocities when the number of image-pair velocities used is high enough (>1000).  The peak in March 2019 is even captured! The performance are however really poor for a number lower than 100, for example in 2014. This is the main limitation of this method.

It would be impossible to retrieve the surge of the Lowell glacier, because only small baseline velocities are available during the surge.

In text, we put: "Hence, TICOI can retrieve monthly velocities using only image-pair velocities with long temporal baselines. It takes advantage of the temporal closure which relies on redundancy of annual velocities, having Sentinel-2 providing new images every 5 days in optimal conditions. However, it still requires a sufficient amount of observations to obtain a reliable time-series (> 500), as also illustrated over the Lower part of the Kaskawulsh glacier (Fig. \ref{fig:KaskL_annual_vel})"

[Figure]

L484-485 this should be 1m and 3m respectively for 0.1pix

Thanks for noticing this. Corrected.

L527 As currently set up this is fully parallelised right? So for ~300 CPU could be run in ~2hr? Seems feasible to run at least all non-ice sheet data with a large machine.

The dask loading is single core and also depends on the I/O speed, so the computation may not scale linearly. However, with large machine with more memory capacity, we could load larger blocks. We need to test it, but we could eventually obtain a lower computation time per pixel. This will make the run of at least all non-ice sheet data even more feasible using 300 CPUs.

We modified the text:

"This computation time remains affordable at the regional scale, and even at the global scale with a large number of CPUs. The computation time could be further reduced, for instance, by taking advantage of GPUs, or by reducing the number of input data by using a stricter outlier filter."

L530 Are there ways to exploit the spatial autocorrelation to reduce computational cost also?

If there is no outlier filter, we could only construct the design matrix A once for the whole cube, but this will not work if the availability of data varies over time from one pixel to another. However, the filtered data is likely to be spatially autocorrelated. If so, we could avoid building some of the design matrix (but it does not seem to work for many pixels when the filter level is strong). If so, the spatio-temporal smoothing could probably be computed on a coarser grid than the velocity grid. We could use the same spatio-temporal smoothed time series to constrain TICOI in a few pixels. However, this could lead to biased results if the spatial autocorrelation is not valid. In summary, we have not implemented these strategies for now, as their potential to reduce computational cost without compromising results is not guaranteed.

L543 'entirely data-driven' – you mean it requires no a priori info/ model here right?

Yes, we add a parenthesis : « TICOI is entirely data-driven (i.e. it does not require strong apriori information on the glacier dynamic) »

L544 KGE improvement can't really be capture in a percentage. From table 1 Kask L had a negative 260% change due to sign flip! Give absolute value increases for it.

We agree with the reviewer, and provided absolute value everywhere. In the conclusion, we wrote : The validation of TICOI results using GNSS data highlights an improvement in RMSE and KGE of around 50\%, and 0.4 respectively in comparison with both the raw image-pair velocities and a rolling median.

L545 I agree this is a useful advance in the way this is done but I probably wouldn't describe it as a paradigm shift. The iterative procedure implemented in GIV can also do this (VWDV and Wickert 2022, https://doi.org/10.5194/tc-15-2115-2021), though almost certainly not as well – the iteration can be rather sensitive to outliers.
On a different note it would be nice to apply slightly more caution in the wording of this. Many areas have limited short baseline data due to high cloud cover (e.g. many HMA glaciers) but also large temporal variability within this period – it probably won't be very effective at reconstructing these. Including an assessment of the Lowell/Kask glaciers in the example above on this might help illustrate.

We agree with the reviewer, and modified the text as:

"It has demonstrated its ability to retrieve monthly velocities using annual image-pair velocities only, when there is sufficient temporal redundancy in the dataset."

We realized during the revision that we forgot to reference GIV in the introduction. Although not requested by the reviewer, we acknowledge that it is an really important work related to this field of research,  and a source of inspiration for our work) therefore, we have added a reference to GIV in the introduction using the term 'iterative weighted monthly averaging'

L555 instead of 'reasonable computational time' using an exact number here would be better.

We added : « Finally, the TICOI workflow offers reasonable computational time for application at the regional scale (0.1 seconds per pixel for large dataset with 80,000 layers in time on 32 CPUs). »

**Reference :**

Charrier, L., Yan, Y., Colin-Koeniguer, E., & Trouvé, E. (2021, July). Fusion of glacier displacement observations with different temporal baselines. In *2021 IEEE International Geoscience and Remote Sensing Symposium IGARSS* (pp. 5497-5500). IEEE.

Zheng, W., Bhushan, S., Van Wyk De Vries, M., Kochtitzky, W., Shean, D., Copland, L., ... & Pérez, F. (2023). GLAcier Feature Tracking testkit (GLAFT): a statistically and physically based framework for evaluating glacier velocity products derived from optical satellite image feature tracking. *The Cryosphere*, *17*(9), 4063-4078

We thank Benjamin Wallis for his careful reading, his detailed comments and help in improving our manuscript. We have made all the changes suggested. Please find below our answers (in black).

Summary:

In this manuscript Charrier et al. present TICOI (Temporal Inversion using Combination of Observations and Interpolation) an open-source package in Python for the post-processing of ice-velocity observations derived from satellite observations. This TICOI package uses the principle of temporal closure of velocity measurements and builds on previous work by several of the authors which demonstrated this technique's application to ice velocity. This manuscript substantially develops this technique by introducing methodological developments to address shortfalls in previous versions, producing an open-source Python package to implement the method, and validating against GNSS glacier motion measurements.

Overall, in my opinion, this is an excellently written manuscript. I found the explanation of the TICOI method to be easy and intuitive to follow with an appropriate level of detail. The presentation of the results is clear, and the performance of the package is impressive, particularly when applied to retrieve sub-annual velocity fluctuations from long temporal baseline measurements. The authors provide a transparent and balanced assessment of the performance of their method, including a comparison to a conventional moving average smoother and using multiple datasets as inputs. I was also pleased to see a thorough discussion of the errors associated with remote-sensing ice velocity measurements, as this is overlooked in many studies.

I am confident that the method and software package described in this manuscript will be of interest to anyone in the Cryosphere community who works with remotely sensed ice velocity datasets. Adoption of this method would improve the velocity products produced in the community, in terms of accuracy and the representation of errors. This has great potential to be valuable to downstream users of these data for applications such as the study of processes influencing ice motion and modelling glacier and ice sheet behaviour.

Additionally, the authors say they will make their TICOI python package available online upon publication of a final manuscript, however, the package is already available at the link provided in the manuscript. Therefore, I took a brief look to assess the quality of the author's python package at this early stage. Even at this point before publication, the TICOI package is well documented including example code. The authors may choose to develop the presentation and documentation of the package further before publication, but as it stands, I have no concerns about the code and data availability. This is an excellent example of how to present open-source code. I applaud the authors for their effort in this regard.

I have a couple of small general comments regarding how the clarity of figures, terminology, and options for shortening the manuscript. After that, I have given line-by-line comments on more specific points. I hope these will be useful for improving the manuscript. See below:

Overall comments:

Figures: The quality of figures in this manuscript is very high. However, there are a few places where I think small changes would significantly improve the clarity and usability of the figures.

The choice of grids and map projection should be made more consistent. In Figure 2, the plots are given with a lat/lon border grid, but Figures 3 and 6 use an x/y grid. I would recommend maintaining the same coordinate system across these plots for better interpretability. Likewise, the units and map projection for Figure 3 are not given in the caption. I assume it is EPSG:3260 like Figure 6, but this isn't clear.

We thanks the reviewer for giving us the opportunity to improve the coherence of the coordinate system used in our Figures. We have modified all the grid to a EPSG:4326 grid. We provide coordinates with a suffix, and 2 significative number. We also specify the map projection in the caption of Fig 3 and 6.

It can be hard to read the figures in this paper when they are presenting large volumes of ice velocity data (eg. Figure 3c, Figure 8a, Figure 11). Could these figures be expanded to the full width of a page, or use vertically stacked separate axes for the image-pair velocities and TICOI results? Similarly, the red/pink/orange colour scheme is difficult to read, especially in Figure 8a. I appreciate that presenting this volume data on one plot is always difficult.

We thanks the reviewer of this suggestion. We splitted Figure 3c in two plots, and increase the size of the two other figures.

Length: The manuscript is somewhat long with 12 figures and 2 tables. Moving some of the figures to the supplementary material could address this and make the paper more focussed on the key results and takeaway messages. Specifically, I would suggest figures 5, 7, 10 and 12.

As suggested by the reviewer, we have moved Figure 5, 7 and 12 to the supplementary material. We kept Figure 10 in the main manuscript to facilitate the interpretation of the spatio-temporal profile.

Use of the term 'regularized': The term 'regularized' is used early on (line 58) here to refer to an even temporal sampling, but later in the manuscript is used for regularization in the context of solving an ill-posed problem. The latter usage is what I would expect the term to refer to in a scientific paper, and I think most readers would approach it this way, too. I think it's okay to use regularized to refer to even temporal sampling where it is clearly explained, like it is with the brackets in line 58. However, in some places in the manuscript it is ambiguous, for example 'regularized' and 'regularization' in the title and abstract could refer to either aspect. For the sake of clarity, the authors could consider choosing a different term to refer to sampling the data on a regular time-step. For example, something like 'temporal standardization'. Although I concede this is not as catchy.

We thank the reviewer for highlighting this point! We have replaced 'regularized' with 'regular' (i.e. sampled at regular time steps) in the manuscript.

Line-by-line comments:

135: There is a hard limit of u =10 for iterations. Can you comment on how often this limit is reached? What is the average number of iterations required?

This limit is rarely reached. The average number of iterations required is about 5 to 6, in our test sites.

156: Could you also comment here on how this RMSE convergence behaves for faster glacier flow, e.g. the > 1000 m/yr speeds that are common for outlet glaciers?

We thanks the reviewer and added : « Note that this asymptote could be reached with a smaller temporal sampling for faster glaciers. »

206: Can you comment on how the VVC metric is affected by changes in ice flow direction? This can be significant on ice-shelves particularly around rifts.

The VVC is only used to select an optimal regularisation coefficient by looking at the inflection point of the VVC curve. If there is a real change in the glacier flow direction (e.g. around a rift, over parts of the ice front or glacier convergence areas), the solution will still converge to a plateau, which has a lower value than for the case of a constant flow direction. In any case, it is advisable to calculate the VVC over relatively large areas (such as a whole massif) in order to be more robust. Lastly, it's important to emphasise that TICOI never restricts the direction of the flow.

We added in the text, at the end of section 2.4:

If there is a real change in velocity direction over time, the VVC curve will converge before 1, but the curve will still have an inflection point. For better robustness, it is advisable to calculate the VVC over a relatively large area.

231: Should be 'validated' not 'validates'

Corrected.

293: In this section, it would be insightful if you were able comment on how the different algorithms perform, ie what are their strengths and weaknesses? They are quite different, so this may be helpful to a reader who is not familiar with these datasets.

We thanks the reviewer for this recommandation and added a few lines :
The strength of ITS\_LIVE is to be available worldwide in open access from 1980s to 2023, while the IGE dataset, published in \citet{millan2019mapping}, covers only two years. The strength of the IGE dataset is its spatial resolution (50 m against 240 m) which allows velocities of relatively small glaciers to be captured. A more detailed comparison is beyond the scope of this paper.

347: I don't think that Figure 3a supports the statement that 'Over stable areas, the difference has median values of 0.0 m/yr'. In Figure 3a it appears that most of the areas outside the glacier outlines have a negative value, as shown by the general blue shading. Can you explain this difference and clarify this point in the manuscript?

We thanks the reviewer for this comment. We have checked our analysis. After reprojecting the raster of differences, the border of the images included nodata with a value of 0. This was taken into account in the analysis. After correcting that the violon graph over the area is:

[Figure]

However, we removed this graph and added a comparison of the statisitcs in stable areas, as suggested by reviewer 2:

Figure 3 caption: The caption refers to white lines for glacier outlines, but they are grey in the figure.

Modified to grey

Figure 3 caption: From the figure and caption it's not obvious to me if the difference being plotted is TICO – TICOI_detect_temp or TICOI_detect_temp – TICOI. Based on Figure 3c I think it's the latter, because this value is positive and point A is red on the map, but it's not clear to me. This should be clarified in the caption.

We thanks the reviewer for spotting this error. The caption was accurate but not the legend of the figures. We are ploting the difference betwen TICOI with and without an automatic detection of temporal decorrelation.

369: In this paragraph including some absolute values as well as % change could be useful for giving the reader a better understanding of the results.

We thanks the reviewer for this suggestion. We added absolute values for RMSE and KGE. We removed the absolute values for the KGE as suggested by the other reviewer.

370: It's confusing to say KGE increased if the range includes a negative percentage.

We modified « an increase in KGE up to 87\%, with a median improvement of 62\% ».

371: In my opinion it is best if the term 'significant' is reserved to refer to specifically to statistical significance. I would consider choosing a different term where possible, also on lines 387 and 445.

We have modified 'significant' and 'significantly' to important, large and drastically.

Figure 9 caption: The caption refers to a logarithmic scale in a), but the colourbar isn't logarithmic.

Thanks for spotting this error.

477: I think this should read 'RMSE even slightly increases'?

We have corrected this error.